# Electrohydraulic musculoskeletal robotic leg for agile, adaptive, yet energy-efficient locomotion

Thomas J. K. Buchner [1,7], Toshihiko Fukushima [2,7], Amirhossein Kazemipour [1], Stephan-Daniel Gravert [1], Manon Prairie[1], Pascal Romanescu[1], Philip Arm [1,3], Yu Zhang [1,2], Xingrui Wang[2], Steven L. Zhang [2], Johannes Walter[2], Christoph Keplinger [2,4,5] ✉ & Robert K. Katzschmann [1,6] ✉

Robotic locomotion in unstructured terrain demands an agile, adaptive, and energy-efficient architecture. To traverse such terrains, legged robots use rigid electromagnetic motors and sensorized drivetrains to adapt to the environment actively. These systems struggle to compete with animals that excel through their agile and effortless motion in natural environments. We propose a bio-inspired musculoskeletal leg architecture driven by antagonistic pairs of electrohydraulic artificial muscles. Our leg is mounted on a boom arm and can adaptively hop on varying terrain in an energy-efficient yet agile manner. It can also detect obstacles through capacitive self-sensing. The leg performs powerful and agile gait motions beyond 5 Hz and high jumps up to 40 % of the leg height. Our leg's tunable stiffness and inherent adaptability allow it to hop over grass, sand, gravel, pebbles, and large rocks using only open-loop force control. The electrohydraulic leg features a low cost of transport (0.73), and while squatting, it consumes only a fraction of the energy (1.2 %) compared to its conventional electromagnetic counterpart. Its agile, adaptive, and energy-efficient properties would open a roadmap toward a new class of musculoskeletal robots for versatile locomotion and operation in unstructured natural environments.

Stationary robotic systems are already indispensable in modern factories, but humanity also desires assistive and collaborative robots that are equally useful in natural and unstructured environments. The success of animals in such environments partly stems from their embodied intelligence; that is, they have evolved in shape and function for given tasks[1]. Forty years ago, the metallic Raibert Hopper[2] introduced legged locomotion as a means of transportation.

Today's legged robots comprise rigid metal structures with discrete links[3–11]. These systems can already walk on uneven terrains (for example, hiking mountain trails)[6], and legged platforms based on electromagnetic drivetrains[12–16] start to surveil security-critical areas or monitor dangerous industrial environments[12]. However, legged systems have not yet reached the agility and adaptivity seen in animals.

[1]Soft Robotics Lab, D-MAVT, ETH Zurich, 8092 Zurich, Switzerland. [2]Robotic Materials Department, Max Planck Institute for Intelligent Systems, 70569 Stuttgart, Germany. [3]Robotic Systems Lab, D-MAVT, ETH Zurich, 8092 Zurich, Switzerland. [4]Paul M. Rady Department of Mechanical Engineering, University of Colorado Boulder, Boulder, CO 80309, USA. [5]Materials Science and Engineering Program, University of Colorado Boulder, Boulder, CO 80309, USA. [6]ETH AI Center, ETH Zurich, 8050 Zurich, Switzerland. [7]These authors contributed equally: Thomas J. K. Buchner, Toshihiko Fukushima. ✉e-mail: ck@is.mpg.de; rkk@ethz.ch

To better mimic animals' versatility, we need to expand the space of materials used to design systems and robots[17–22]. Designs inspired by the musculoskeletal architecture of animals, such as bones, contracting actuators, and tendons, have been employed to enhance robotic performance[23–26]. This biomimetic design not only amplifies a robot's versatility and adaptability[23,27,28] but also facilitates the control of high-degree-of-freedom (DOF) systems with simpler controllers[29,30]. In pursuit of musculoskeletal robots, recent research has been focused on electromagnetic motors[31–33]. While electromagnetic motors with rotational encoders are precisely controllable and can produce high torque, they tend to be bulky and present low compliance and back-drivability. Although ongoing research aims to improve these limitations[34], electromagnetic motor-driven systems require complex controls to "approximate compliance" through software—the control loop speed limits software-based compliance, resulting in reduced safety during accidental fast impact. Direct driving transmissions provide back-drivability to the system; however, these actuators continuously consume energy while holding a payload unless they are fitted with complex clutch systems[35].

We therefore hypothesize that musculoskeletal robots powered by electrohydraulic artificial muscles[36,37] may open a roadmap to agile, adaptive, and energy-efficient robots for versatile locomotion and operation in natural environments (Fig. 1). Conventional electromagnetic motors with rotational joint encoders are therefore replaced with antagonistic pairs of artificial muscles that contract linearly and can self-sense their contraction state. Drivetrains in joints are also

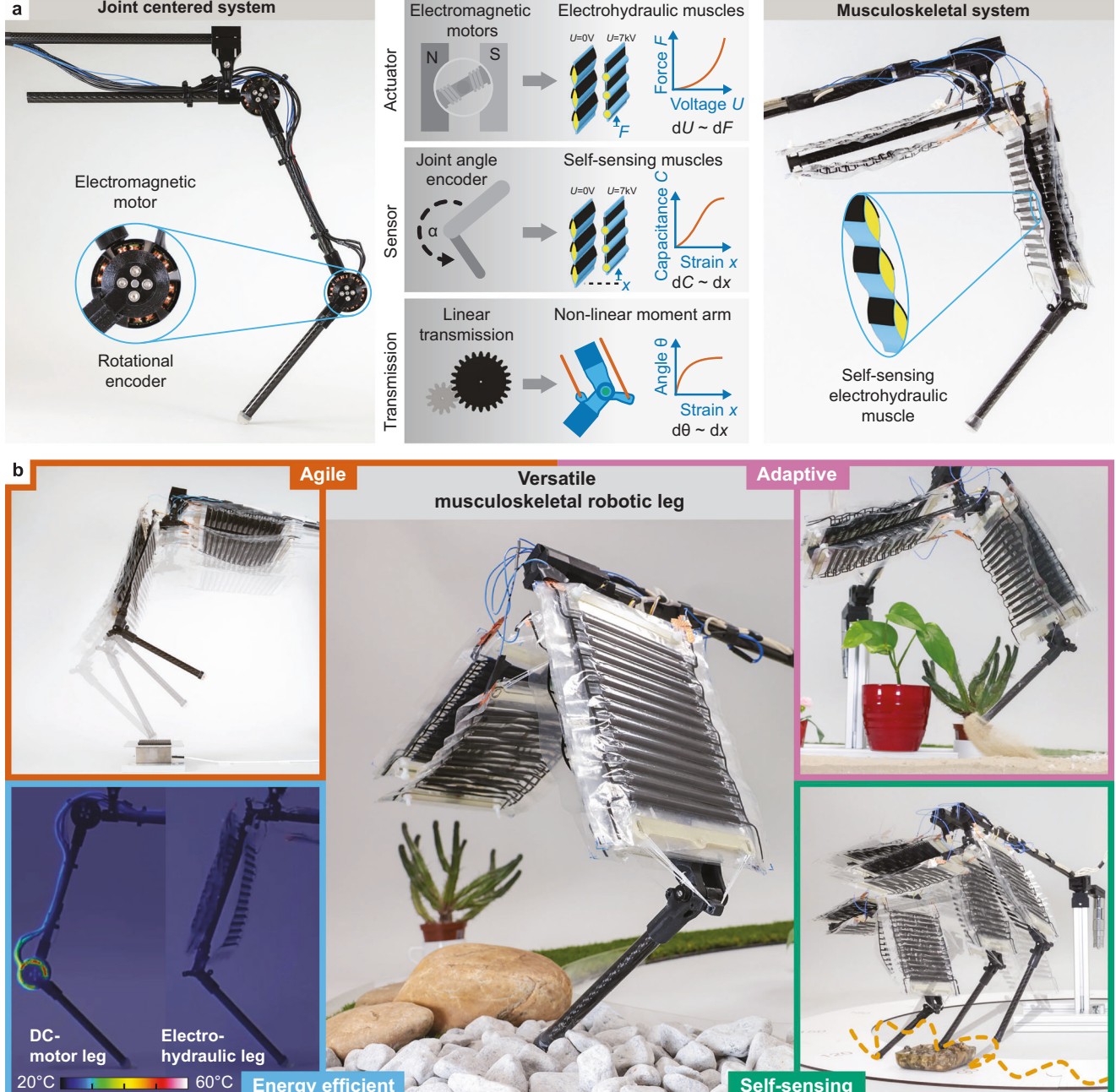

**Fig. 1 | A robotic leg built with electrohydraulic artificial muscles enables versatile robots. a** Conventional robotic legs consist of electric motors, position encoders, linear transmissions, and a rigid base structure. The motor and encoder can be replaced by electrohydraulic artificial muscles that are antagonistically placed. A change in voltage applied to such muscles leads to forces and strains, whereas the strain can be self-sensed by measuring the capacitance of the muscle system. **b** The versatility of the leg is shown with experiments testing agility, adaptivity, energy efficiency, and self-sensing.

replaced by a bioinspired tendon routing over a joint, which creates a nonlinear moment arm and results in a suitable torque output of the contracting muscle (Fig. 1). We test the hypothesis by introducing PELE, an agile and energy-efficient Peano-HASEL (hydraulically amplified self-healing electrostatic actuator) driven leg with 2-DOF. We use the leg isolated from complete robotic systems because the leg is a crucial component in legged robotics to study the impact of actuator choice[38,39]. Here, antagonistic pairs of electrohydraulic muscles[36,37] are connected via tendons to a lightweight carbon fiber skeleton.

We test PELE in a range of locomotion benchmarks and eventually find that it can dynamically adapt the stiffness of its muscle packs, allowing it to hop over various terrain in an agile yet energy-efficient manner (Fig. 1b). To demonstrate the controllability of the leg, a closed-loop PID controller and encoders allowed the leg to track trajectories in task space accurately. Moreover, the musculoskeletal design and the electrohydraulic actuators of PELE enabled an open-loop force controller via only regulating applied voltage to manipulate and locomote the leg. PELE also performed powerful and agile gait motions beyond 5 Hz, linear motions beyond 10 Hz, and high jumps up to 40 % of the leg height (Fig. 1b top left). PELE also exhibited tunable stiffness via voltage regulation and inherent adaptability, which allowed it to hop over grass, sand, gravel, pebbles, and even larger rocks using only a single set of open-loop force control commands (Fig. 1b top right). PELE is also efficient, with a low cost of transport between 1.79 and 0.73 depending on the type of locomotion. The leg is thus more energy-efficient than most systems based on electromagnetic motors. PELE also requires only ~1.2 % of the energy a comparable DC-motor-driven leg would need when performing a squatting experiment (Fig. 1b bottom left). We utilize capacitive self-sensing capabilities[36,40] to let PELE detect and subsequently overcome obstacles by switching its actuation mode (Fig. 1b bottom right). The leg can hop on varying terrain using its inherent proprioception without relying on knowledge of the joint angle (for example, encoder values).

## Results

In the following, we present the details of our results on the design and control of the legged platform (Fig. 2). We highlight the experimental results demonstrating agility, adaptivity, energy efficiency, and self-sensing. Figure 1 and Supplementary Movie 1 give an overview of all results.

### PELE leg

PELE consists of a carbon fiber backbone with 3D printed joints (hip and knee) and two sets of electrohydraulic artificial muscles attached to it via tendons (Fig. 2a). The muscle packs consist of parallel stacked electrohydraulic muscles and a tendon attached at the end of the pack. The leg has four muscle packs and is driven by up to four separate high-voltage amplifiers controlled via a computer.

Each electrohydraulic muscle is a Peano-HASEL comprising serially stacked actuator pouches[40]. A single pouch is a polymer shell filled with liquid dielectric and covered with electrodes on either side. When different electric potentials are applied on the two electrodes of the pouch, charges are moved to the electrode, and electrostatic forces cause a shape change in the pouches. This shape change leads to a linear contraction $\Delta x$ along the serially stacked actuator pouches. The contraction is reversed once the electrodes are discharged. Electrohydraulic actuators usually have catch states, where no additional energy is required to hold a position, even under load[41,42], except for a tiny amount of charge leakage through dielectric layers that must be compensated. Therefore, PELE has very little power consumption while holding a posture, even when exerting a substantial joint torque (Supplementary Table 1).

In contrast to electromagnetic motors, where supplied current correlates to actuator torque output, the voltage applied to an electrohydraulic actuator correlates to actuator force. The controlled actuator force output, combined with the antagonistic pair of muscles and the actuator's force-strain characteristics, allows the leg to locomote in open-loop force control mode without requiring a joint angle encoder. PELE is, therefore, inherently adaptive. The tendon connected to one end of each muscle transmits the muscle's force and change in length. For PELE, we attach one end of the actuator rigidly to the leg and the tendon on the other end to the shank after the respective joint (Fig. 2a and Supplementary Fig. 1). Each tendon has a nonlinear moment arm transmission for suitable angle-torque profiles of the joints (Supplementary Fig. 2). We define a Cartesian coordinate frame for the robotic leg at the rotational axis of the hip joint of the leg. The inset in Fig. 2a details the coordinate frame and the definition of the angles for the hip and knee joints.

The range of motion of PELE can be observed when we apply voltages to the individual muscles. Here, we contracted the muscle packs in repetitive cycles at 0.5 Hz. Each position relates to a specific set of voltages applied to the four muscle packs of PELE (Fig. 2b).

We utilize a cascaded closed-loop controller (Supplementary Figs. 3, 4) to command the tip position of the leg precisely. Magnetic encoders are embedded in each joint to measure the joint angles. The angles are converted into relevant HASEL actuator displacements. We successfully tracked various shapes with the tip of PELE over cyclic trajectories for a duration of 20 s (Fig. 2c and Supplementary Movie 2). In addition to precise position control, the leg enables agile and fast locomotion, which will be further discussed in the following section.

### Locomotive agility

In this section, we demonstrate the dynamic and agile motions that our leg can achieve. The leg's low inertia, combined with the muscles' fast response (Supplementary Fig. 5) and high power-to-weight ratio (Supplementary Fig. 6), enabled the leg's agile motions. Supplementary Movie 3 shows experiments that demonstrate the leg's agile locomotion.

The leg achieved a jump height of 128 mm while maintaining a short stance time of 91 ms. This showcased the leg's ability to perform highly agile jumping maneuvers (Fig. 3a). The short stance time indicates the short charging time of the muscles and the low moment of inertia of the system. Moreover, the leg's vertical jumping agility[43] ($V_{jump}$, Eq. 3) was 0.75 m/s, and the jump frequency[43] ($F_{jump}$, Eq. 4) was 5.8 Hz.

Given the short stance time, we were encouraged to explore the leg's capabilities of rapid hopping, that is, high-frequency repetitive jumps (Fig. 3b). The gravitational acceleration alone was not high enough to bring the leg back down to the ground fast enough, thus limiting the achievable maximum motion frequency. To account for this in our experiments, we used the antagonistic pairs of muscles in the leg to reach even higher frequency motions. Using an open-loop force controller (Fig. 3b; top), we successfully achieved a hopping frequency of 3 Hz on various terrains (Fig. 3b; bottom), showcasing the leg's agility and versatility. The hopping motion displayed stable limit cycles (Fig. 3c), further emphasizing the leg's ability to perform robust and agile hopping maneuvers. For 3 Hz, the leg achieved a jump height of 80 mm and was dynamically stable in vertical hopping.

We also investigated high-frequency motion patterns without ground contact. We employed an open-loop force controller with phase-shifted sine signals for each muscle ($V_{in}$, Eq. 5, Fig. 3d; top) to achieve higher frequency gait motions (Fig. 3d; bottom). Without ground contact, the leg demonstrated a running motion with a frequency of 5 Hz and a linear motion with a frequency of 10 Hz (Fig. 3d; bottom). This high frequency exemplifies the leg's ability to perform rapid and agile gait cycles. The results indicate that antagonistic pairs of muscles can inherently generate human-like gait motion trajectories[44]. We also observed that the foot's range of motion changed for different operation frequencies, with a maximum of 3 Hz (Fig. 3e), which matched the natural frequency of the leg system.

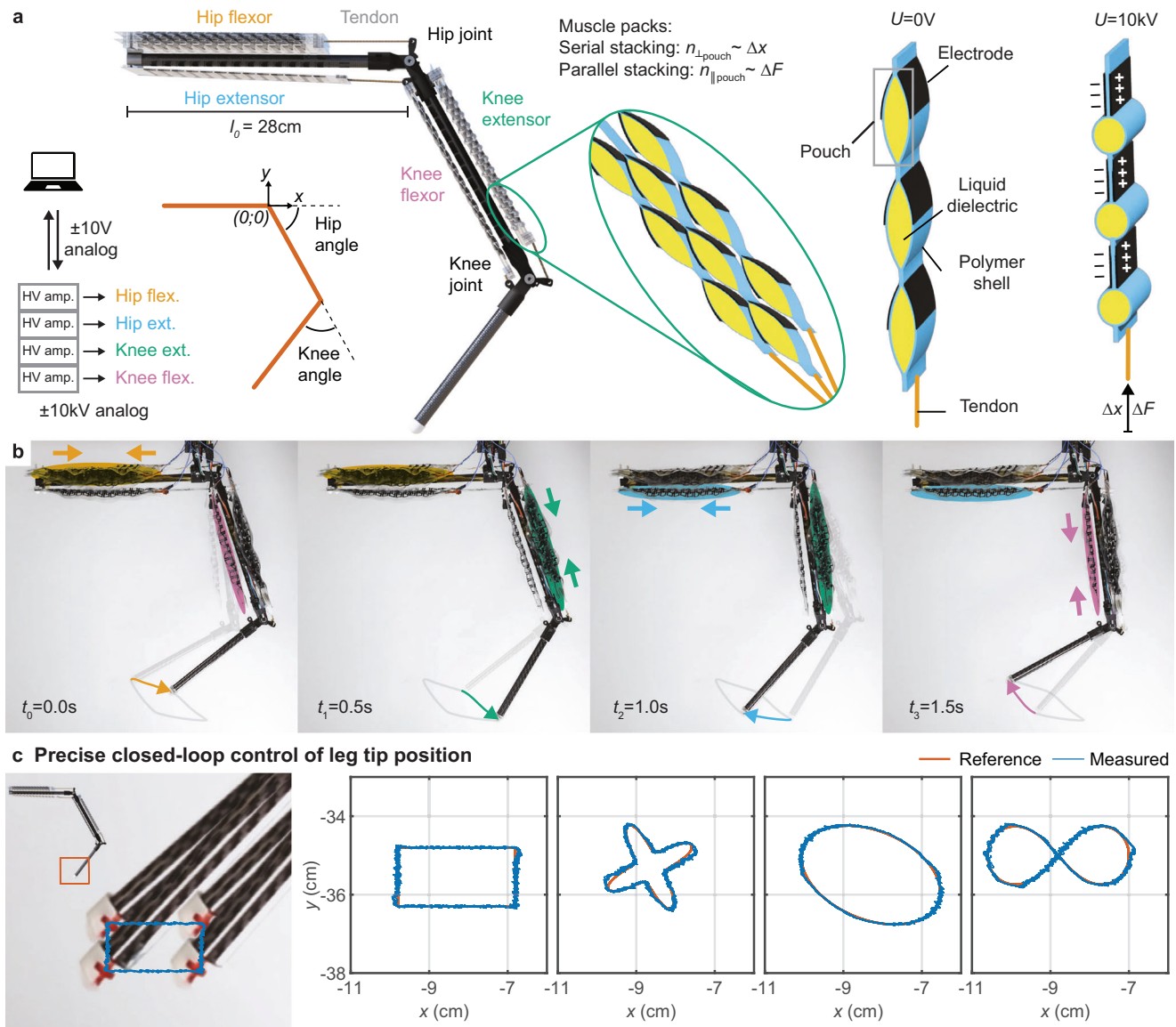

**Fig. 2 | Electrohydraulic artificial musculoskeletal leg system. a** System schematic of our leg. A computer interfacing with four high-voltage amplifiers, one for each muscle pack, i.e., hip flexor and extensor as well as knee flexor and extensor. Each muscle pack consists of parallel stacked actuators comprising serially stacked pouches. A pouch contracts linearly under the application of a voltage. **b** Working mechanism of PELE. We applied a voltage to the muscle packs at a frequency of 0.5 Hz, leading to a cyclic motion of the leg's tip position. **c** Closed-loop position control of the leg's tip position. The leg's tip position (relative to the origin located at the leg's hip joint depicted above) was close-loop controlled to precisely track the trajectory of four predefined shapes for 20 s. Encoders in each joint translated angle measurements into actuator displacements.

We successfully tested the leg in several jumping, hopping, and in-air motion experiments. The results collectively showcase the leg's versatile and agile capabilities in high jumping, multiple rapid hopping, and fast gait cycles.

**Inherent adaptivity**

The robotic leg, enabled by back-drivable muscles, displayed an inherent adaptivity to changes in terrain. With a feed-forward force controller, which cyclically generated one single actuation pattern of the applied voltage (Fig. 4a), the leg was able to hop over diverse natural terrains, such as stones, sand, gravel, and grass (Fig. 4b). This experiment aimed to demonstrate the leg's adaptability to different terrains. The leg successfully jumped through the various terrains, showcasing its ability to adjust and adapt dynamically to the changing ground conditions. In this experiment, the leg was driven by a single feed-forward force controller that employs cyclic voltage signals. The adaptive behavior on each terrain emerged through interactions between the leg and the environment. Supplementary Movie 4 shows these experiments on adaptive locomotion. The leg's inherent adaptive behavior contrasts conventional DC-motor-driven systems that require complex controllers to simulate adaptivity.

We investigated the adaptability of the leg's locomotion across different terrains, transitioning between soft (sponge) and hard (tabletop) surfaces (Fig. 4b). The leg exhibited a cyclic hopping gait on a sponge surface, forming a stable limit cycle (Fig. 4b; inset). When the sponge was removed, the leg first entered a brief, chaotic period and then seamlessly transitioned from a stable hopping gait on the sponge to a new stable hopping gait on a tabletop surface (Fig. 4c). Notably,

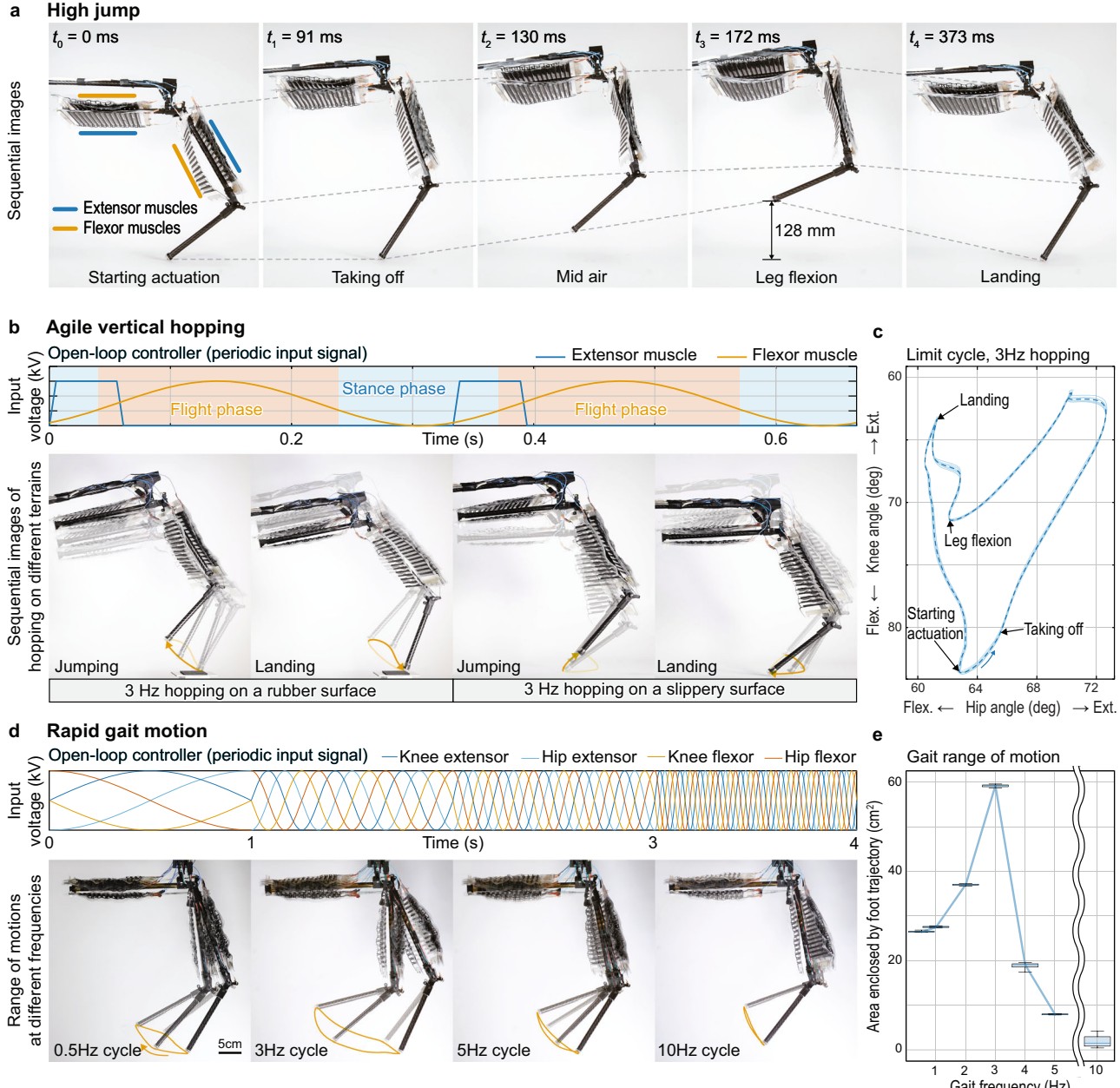

**Fig. 3 | The leg performs agile locomotion. a** High jump. The leg jumped 128 mm high from the ground and reached the highest point in 172 ms. **b** Agile vertical hopping. Top: the leg was driven by an open-loop force controller with the periodic input signal. The partial overlap of the activation of the extensor and flexor muscles during the stance phase was used to compensate for dead zones of the muscle (from 0 kV to 3 kV, depending on external loads, the actuators do not respond to voltage signals), and for inertial delay of the leg motion. Bottom: the leg hopped at 3 Hz on the rubber surface (left) and the slippery surface (right). **c** Limit cycles for hopping on a slippery surface. The dashed line showed the mean value of the 15 cycles. **d** Rapid gait motion. Top: the leg was driven by the open-loop force controller with the periodic input signal. Bottom: The leg achieved 5 Hz gait motion and 10 Hz linear motion. **e** Size comparison of the area enclosed by the leg's foot trajectory at each frequency. The mean and standard deviation were calculated from multiple motion cycles after reaching a steady limit cycle (details in the Methods section).

this transition occurred inherently without any change in control input.

Additionally, we investigated the leg's ability to achieve a soft landing through the muscle's back-drivability. The leg's joint angles were recorded as it was dropped from the same height onto a surface using different constant actuation voltages. The leg achieved soft landings just by maintaining a fixed voltage (Fig. 4d). The results also revealed tunable stiffness of the leg where lower actuation voltages led to softer landings with substantially higher changes in joint angles (Fig. 4e) and larger compressive bouncing (Fig. 4f). This inherent soft-

landing feature is enabled by the muscle's inherently tunable back-drivability (Supplementary Fig. 7) and eliminates the need for complex computational controllers[45] to adjust the leg's stiffness.

## Energy-efficient locomotion

In this section, we present two experiments to demonstrate the energy-efficient locomotion of the leg (Supplementary Movie 5). First, to showcase the high energy efficiency of locomotion, we evaluated the net cost of transport (COT, Eq. 6) during locomotion (Fig. 5a, b). We specifically investigated two different types of gaits: hopping gait

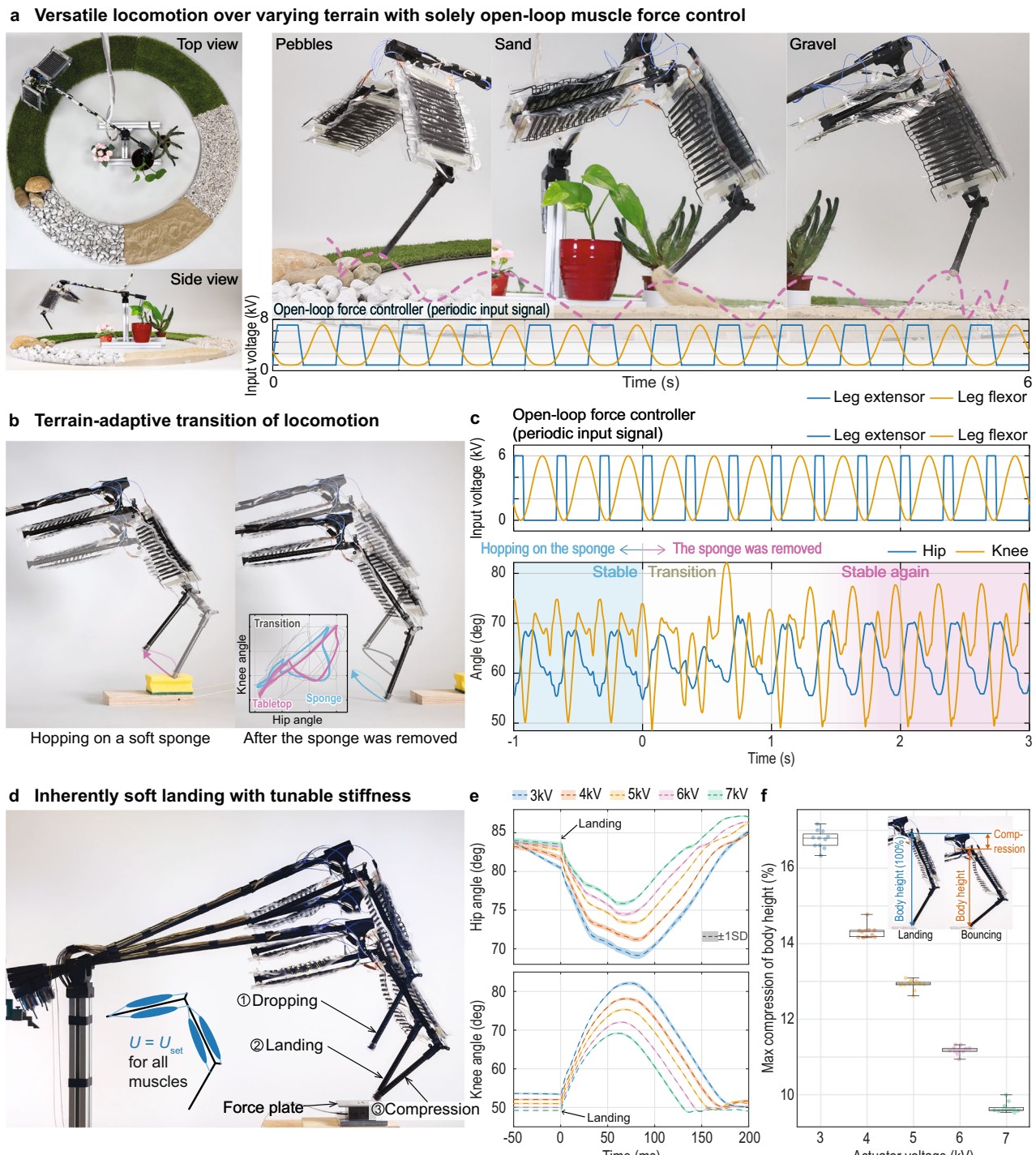

**Fig. 4 | The leg is inherently adaptive using open-loop force control. a** Versatile locomotion over varying terrain with solely open-loop force control. **b** Terrain-adaptive transition in locomotion. Left: The leg hopped stably at 3 Hz. Right: While hopping, the sponge was removed, and the leg adaptively transitioned onto the new surface and hopped stably again. Inset: Transition of the two types of limit cycles. **c** Top: periodic input signal. Bottom: Knee and hip angle over time. The leg was hopping stably on the sponge terrain (blue shaded area), and after leaving, the leg's cycle converged to another stable cycle (pink shaded area). **d** Inherently soft landing with tunable stiffness. The leg was dropped while maintaining a set voltage on all muscles and showed a soft landing. **e** Hip and knee angle change during the compression period. The angle changed less as the voltage increased. The standard deviation from the average for each voltage ($N = 11$ per voltage) is shown in semi-transparent for each voltage measured. **f** Compression ratio along the voltage levels. Higher voltage made the leg stiffer on landing. The individual data points ($N = 11$ per voltage) and the average and standard deviation are shown in a box plot.

and crawling gait. We defined crawling gait as hopping without a flight phase (it can be seen in Supplementary Movie 5 from 00:28 to 00:35). We investigated the influence of the leg's mass on its energy efficiency. Therefore, we considered two scenarios: first, the potential reduction of the robot's mass achieved by further light-weighting of the actuators

as detailed in ref. 46, and second, the increase of mass when adding mobile power supplies and batteries for untethered operation in the future[47–49]. To test these scenarios, we adjusted the counterweight mass on the boom arm to emulate the robot's mass at five levels from 198 g to 391 g (−27 g to +166 g).

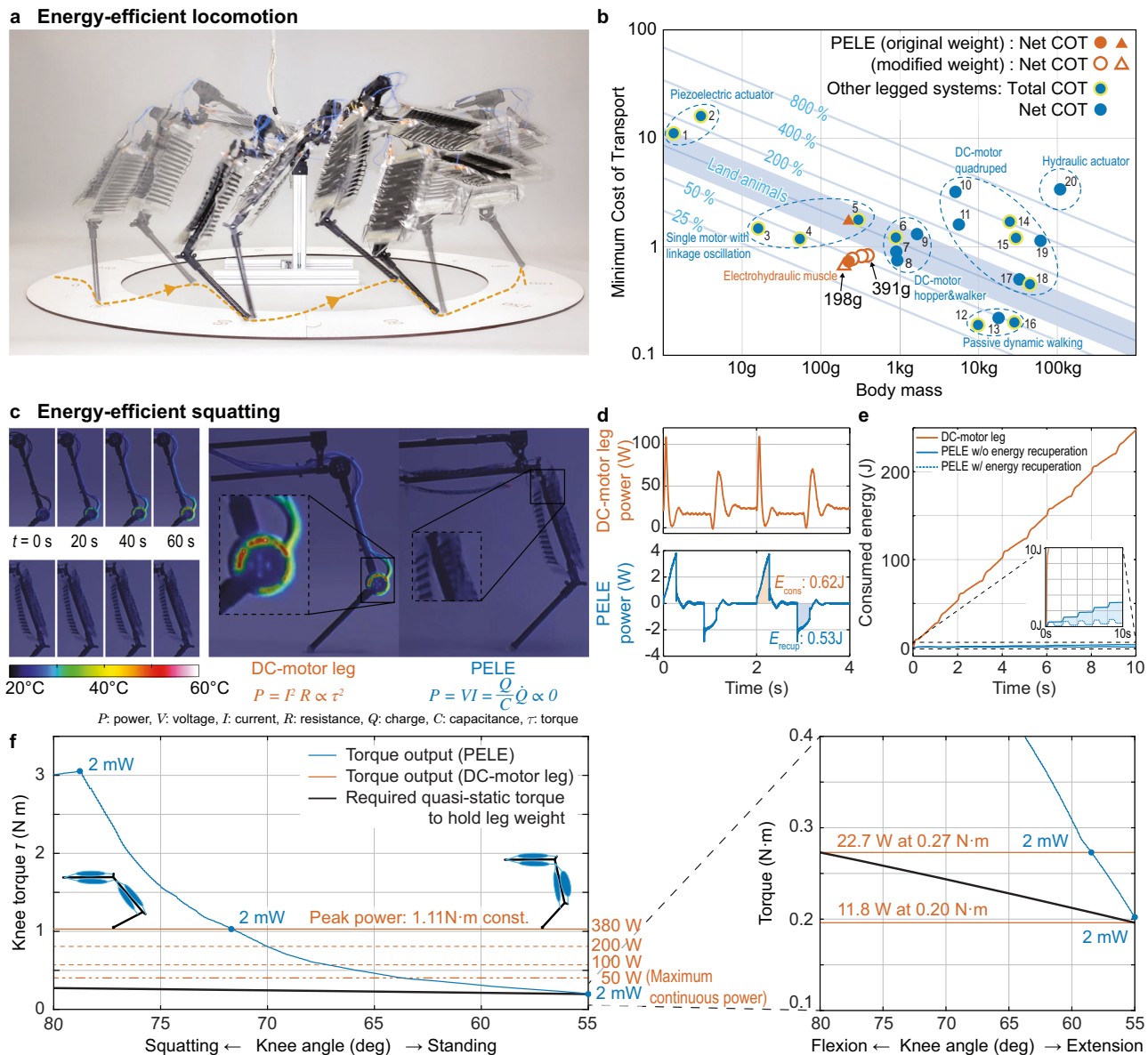

**Fig. 5 | Energy-efficient locomotion. a** The leg hopped forward using an energy-efficient gait. **b** PELE's minimum net cost of transport (COT) compared to other legged systems. PELE performed a hopping (triangle) and crawling (circle) locomotion. We detached and attached counterweights to the boom arm to investigate the effect of weight change. As a general reference, the 100% COT line indicates the metabolic cost of land animals[60]. Total COT includes all the power consumption on the robot (including computers, drivers, cooling fans, etc.). Net COT accounts only for the power consumption of actuators. PELE's mass (225 g) was varied from −27 g to +166 g. The hopping mode was most efficient for the lower weights, and the crawling mode was most efficient for the larger weights (COT in Supplementary Table 2). **c** Thermal images of PELE and a DC-motor-driven leg during squatting motion. PELE was compared with a DC-motor leg of comparable weight and torque output. The image shows no observable thermal energy loss for the HASEL actuator. **d** Comparison of power consumption during squatting. PELE showed the potential for energy recuperation from the gravitational potential energy. **e** Comparison of the consumed energy over time. For PELE, an area is shown where the lower line indicates the energy consumption under perfect recuperation, and the upper line indicates no recuperation. **f** Analysis of the quasi-static torque needed to hold the leg in different positions (knee angles). Orange lines indicate the DC-motor's torque and power consumption. The blue line indicates the torque for PELE's actuators at different knee angles.

The results indicate that the leg on a boom arm achieved a desirable low cost of transport, reflecting efficient movement with low energy consumption. Notably, the leg achieved a minimum net COT of 0.73 at the original weight of 225 g and a net COT of 0.69 and 0.83 at a reduced weight of 198 g and an increased weight of 391 g, respectively. Intriguingly, as the weight exceeded 225 g, the locomotion type that yielded the minimum net COT transitioned from hopping to crawling. Compared to other legged systems (Supplementary Table 3), PELE exhibited remarkably favorable COT values within this range of robotic mass (Fig. 5b and Supplementary Table 2). The net COT calculation does not include the DC-DC high voltage conversion efficiency in driving electronics, typically around 75%[50,51]. These findings suggest that the electrohydraulic muscle system has the potential to enable highly efficient locomotion in untethered, legged robots.

We compared PELE and a brushless DC-motor-driven leg to investigate the underlying factors contributing to efficient locomotion. Details on the DC-motor leg can be found in the Methods section. Both legs performed a cyclic squatting motion, and measurements were taken for temperature change and power consumption. The results revealed a strong contrast between the two leg types. The DC-motor leg exhibited a substantial temperature change, indicating higher power consumption. In comparison, PELE showed no

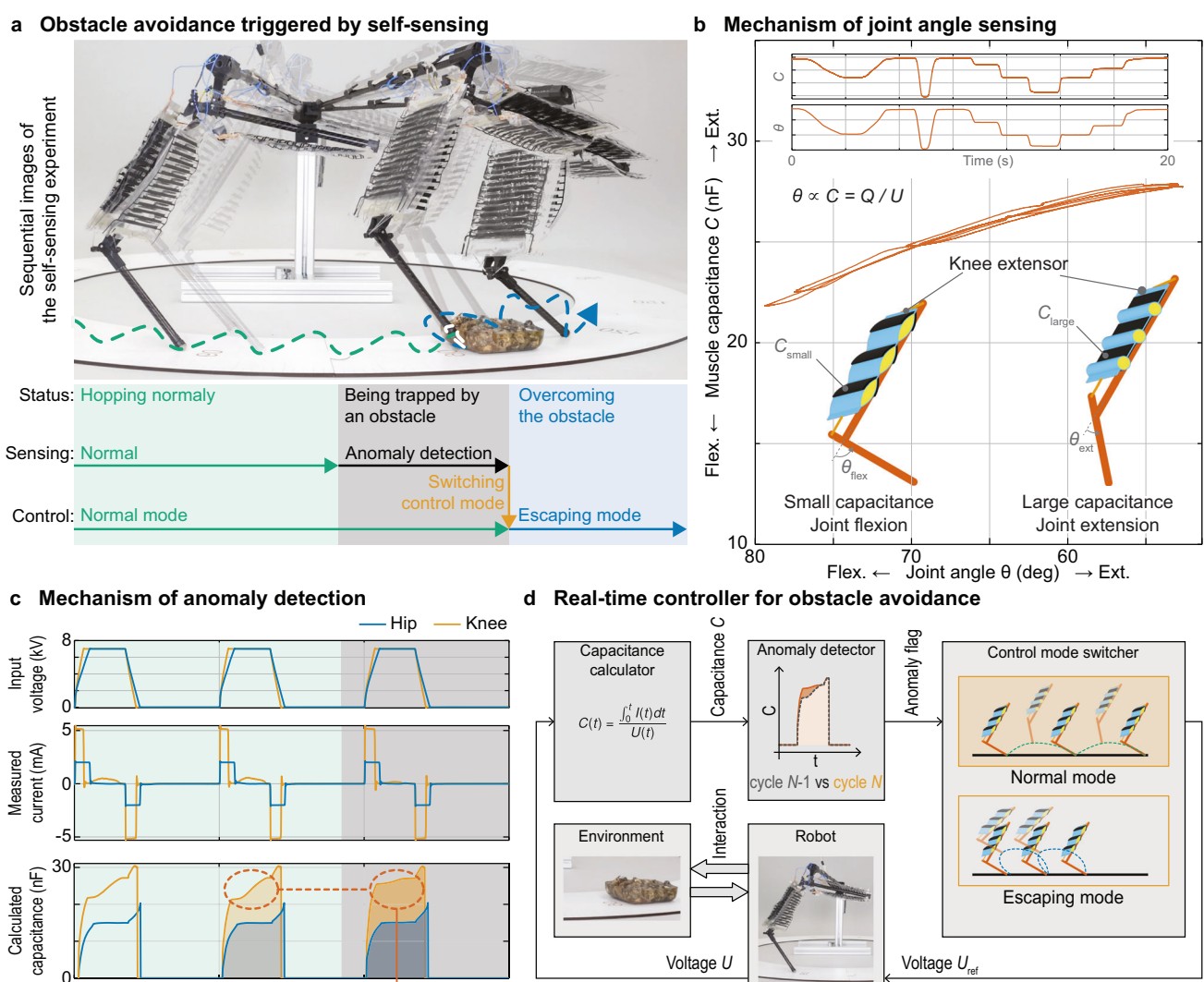

**Fig. 6 | Inherent self-sensing for detection of collisions with obstacles.**
**a** Sequences of obstacle avoidance triggered by self-sensing. Left: Using the normal mode, the leg hopped at 3 Hz and quickly gained distance. Center: Once an obstacle entrapped the leg, the anomaly detector swiftly detected the obstacle. The controller dynamically switched to the escaping mode, reducing the hopping frequency to 2 Hz while bending the knee and pulling the hip forward in the air. Right: The leg successfully freed itself from the obstruction. **b**, The capacitance value of the muscle at a particular joint angle. The leg maintained a constant voltage, while only an external force changed the joint angle. **c** A change in the calculated capacitance pattern given a supplied voltage and current allows for detecting an obstacle collision. **d** Controller diagram for real-time obstacle avoidance. For the leg to hop, the controller employs periodic feed-forward signals as the output voltage to the leg while continuously calculating the knee extensor muscle capacitance $C$. The controller compares the current capacitance values with the values from the previous cycle to detect an anomaly caused by an obstacle collision. Once a collision is flagged, the controller switches to escaping mode to facilitate the leg's recovery.

observable temperature change (Fig. 5c). This finding on thermal heat loss suggests that the electrohydraulic muscles consume very little power (Fig. 5d, e).

In contrast to the DC-motor leg's average power consumption $\bar{P}_{cons}$ (Eq. 8) of 25.3 W during the squatting task, PELE consumed 306 mW, 1.2% of the DC-motor leg. Additionally, power consumption data indicated that PELE consumed only 9.4 mW to maintain its elevated position, while the DC-motor leg consumed 17 W.

Please note that we used a direct-drive architecture for the benchmark DC-motor leg. State-of-the-art DC-motor-driven legs with higher transmission ratios are likely more efficient than a direct-drive DC-motor leg, and the DC-motor leg could be optimized for specific squatting tasks. Nevertheless, our selection of the direct-drive approach was aimed at matching the mass, back-drivability, bandwidth, and simplicity of the actuators used to drive PELE. This design choice results in the versatility of the DC-motor leg being used in

different tasks, mirroring PELE, where a single design achieved all the various tasks. For the DC-motor (T-MOTOR, MN4006), the selection criteria were its weight and its output torque; its weight was comparable to that of the Peano-HASEL stacks and its output torque met the quasi-static joint torque requirements (as illustrated in Fig. 5f). Notably, the power consumption of the DC-motor escalates with the square of the output torque (orange lines in Fig. 5f). By contrast, the Peano-HASEL exhibits constant, minimal power consumption across all torque levels while holding a posture (blue line in Fig. 5f).

Moreover, when PELE returned to its original posture, a significant part of the potential energy gained from the leg standing up was sent back to the power supply (Fig. 5d). In the future, there is the potential to recuperate this energy coming back from the actuators through suitable driving electronics that yet must be designed. Assuming we can capture all the energy from discharging the Peano-HASEL actuators, the average power consumption could reach a minimum of

43 mW ($\bar{P}_{ideal}$, Eq. 9) instead of the 306 mW reported here. We calculated the energy recuperation potential ($E_{RP}$, Eq. 10) of PELE to be 85%. The $E_{RP}$ for the DC-motor leg was negligibly small for the motions commanded during the experiment, especially considering the motor's significant losses due to heat. These results demonstrate the potential of energy-efficient locomotion utilizing electrohydraulic artificial muscles in mobile, legged robots.

## Self-sensing for obstacle detection

We demonstrate the leg's self-sensing ability through an obstacle detection test. The leg was controlled with a real-time controller when it successfully self-sensed an obstacle collision during hopping without any external vision, haptics, or angular sensors (Fig. 6a, Supplementary Movie 6). The capacitance value of the HASEL-type muscles was used to represent the muscle length during their activation (Fig. 6b), as established in previous studies[52–54]. Leveraging this self-sensing function, we designed a controller for collision detection (Fig. 6c, d). The controller employed cyclic feed-forward signals as the output voltage to the leg, enabling it to hop while continuously calculating the capacitance values C at each sampling time. By comparing the current capacitance values with the values from the previous cycle, the controller could identify a certain amount of deviation, signaling a collision with an obstacle. Once a collision was detected, the controller switched the control mode to escape the obstacle, facilitating the leg's recovery.

During the experiment, the leg initially hopped cyclically in the normal distance mode, where it hopped at 3 Hz to gain distance rather than height before encountering the collision (Fig. 6c; green). Subsequently, the leg became stuck at an obstacle (Fig. 6c; gray), and the collision detection approach promptly identified the collision. After a few cycles of the leg being stuck, the controller dynamically switched the mode into escape mode. In this mode, the leg's hopping frequency was reduced to 2 Hz, and the knee was bent while the hip was pulled forward in the air, enabling the leg to successfully free itself from the obstruction.

The successful demonstration of the leg's self-sensing ability and the efficacy of the control approach for collision detection highlights the leg's potential for use in situations where the robot cannot use joint angle encoders (for example, a multiple DOF joint such as a shoulder joint).

## Discussion

PELE showed that (i) electrohydraulic HASEL actuators are functional artificial muscles for a new generation of robotic systems. Our leg's musculoskeletal architecture and the compliant and soft nature of the electrohydraulic actuators allowed for versatility in various experiments.

The leg can (ii) inherently adapt to various types of rough and soft terrain and hop over it using solely open-loop force control; typically, legged robots require complex closed-loop control to achieve this task. The system's compliant antagonistic design has the inherent advantage of being resilient to external disturbances, which minimizes the need for additional sensing capabilities or closed-loop control. This result indicates that musculoskeletal system architectures based on antagonistic pairs of muscles present an avenue for developing new types of bioinspired control and learning strategies.

Our experiments on energy efficiency (iii) showed the possibility for a substantial reduction in energy usage of robotic systems built with this class of actuators, particularly for motion sequences that require holding torque constant for specific periods in a squatting experiment; our design only required ~1.2% of the energy a comparable DC-motor-driven leg would have consumed. Electromagnetically driven setups often require cooling fans not to overheat, especially under such static loading scenarios. The lack of heat build-up under static loading of PELE is also advantageous for applications in thermally sensitive environments.

PELE (iv) showed agile gait motions at over 5 Hz, allowing rapid jumps of more than 40% of the leg height. This high level of agility indicates potential applications of such legged systems for tasks that require highly dynamic and powerful motion. These results show that PELE already reaches levels of agility, which is currently not possible with most electromagnetic leg designs. Yet, substantial performance improvements in specific power and energy are foreseeable for electrohydraulic HASEL actuators, which will decrease the weight of the leg while improving strength and agility[46,55].

We (v) showed that the self-sensing capabilities of HASEL actuators[36] allow the leg to change its actuation pattern when encountering obstacles. This adaptive behavior is achievable using only the actuators' inherent self-sensing abilities, all without incorporating dedicated pose-sensing mechanisms into the leg's closed-loop control system. These self-sensing capabilities reduce the need for additional sensors and allow for control strategies that mimic the proprioceptive capabilities of animals.

The HASEL actuators have a lifetime of thousands up to exceeding millions of cycles[56], depending on the design and demand of a particular application. In the current design, the actuators are not shielded from environmental contact. In a real-world application of a HASEL-driven system, an enclosure or puncture-proof layer (for example, aramid fibers) around the actuator would be necessary.

The main limitations of our PELE presented here are fivefold: (i) The leg is tethered to a stationary control unit and power supplies that can readily provide arbitrary high voltage signals (in the range from −10 kV to 10 kV) at desirable power (≥50 W) levels. Further research into compact and lightweight multi-channel power supplies with sufficient power output is required to make this class of robots untethered while keeping them agile. (ii) So far, we have only shown a single 2-DOF leg mounted on a boom, which limits the leg's locomotion to a circular path. Hexapedal, quadrupedal, and bipedal systems must be investigated to characterize the full potential of this design architecture. (iii) We only compared the energy efficiency for squatting. It would be essential to extend the investigation into other types of motion, such as continuously fast actuation like those used for running. (iv) While it is possible to recuperate a large amount of electrical energy when discharging the actuators, driving electronics incorporating such charge recuperation have yet to be developed. (v) Currently, the electrodes of the actuator are open to the environment and at a high voltage potential. High voltages alone are not necessarily dangerous, only when paired with high currents. HASEL actuators operate at high electric fields (high voltages) and low currents, within established safety standards, unless we perform highly dynamic motions like jumping. Additional safety measures would be required to ensure safe operation in practical applications outside a controlled laboratory environment. These could involve (a) implementing mechanisms that detect electrical breakdowns and fault currents and trigger fast shutoff circuitry for driving electronics and (b) limiting maximum capacitance/energy stored in the actuator system. Additionally, (c) introducing electrical insulation layers[47] to prevent physical contact with the electrode and, therefore, electrical shock can help mitigate such risks[57].

To conclude our original hypothesis, we can confirm that our proposed musculoskeletal robotic leg architecture is agile, adaptive, and energy-efficient in various versatile locomotion tests, even across varying terrain. Therefore, the architecture will serve as a platform for other researchers to build upon in the future. Its full potential will become apparent once a bipedal or quadrupedal version successfully operates in the wild in an untethered, energy-recuperating, and self-balancing manner. Considering the results presented here, this musculoskeletal design paradigm will enable future robots to navigate

complex environments more efficiently and help us come closer to the agility and adaptivity currently only seen in animals.

## Methods

This section first presents the robot's system design, including the skeleton's and the muscle's architecture and the mechanism for self-sensing and transmission. Next, the three types of experimental setups are detailed. Finally, the core analysis methods for these experimental setups are explained.

### Robot system design

In this section, we describe the robot's design in detail. We describe the skeleton's architecture, muscles, and self-sensing mechanism. We also detail the transmission mechanism linking the skeleton and the muscles.

### Skeleton architecture

The skeleton of the robotic leg was constructed from a cut-to-length carbon fiber tube, incorporating 3D printed parts for the joints and mounting adapters, machined plastics for low-friction bearings and shafts, and custom laser-cut mounts for attaching the actuators. The skeleton architecture is apparent from Fig. 1. The structure was iteratively improved to reduce the weight while considering the rigidity and functionality of the system. A lightweight joint setup was manufactured using 3D printing of polylactic acid (PLA). The joint was adhered to the carbon fiber tubes. The bearing and shafts within the leg's joints were machined from polyoxymethylene (POM) and exhibited low-friction and self-lubricating characteristics. The artificial muscle packs were attached at one end to the leg using custom laser-cut Poly (methyl methacrylate) (PMMA) mounts and nylon screws. The other end was connected to the next section of the leg via tendons made from a low-stretch, high rupture strength multi-filament fishing line. This tendon provided a robust and reliable connection between the actuators and the leg. To achieve high performance, the muscles and tendons within each pack were brought to the same length to minimize misalignment. This adjustment ensured that the muscles of one pack could simultaneously contract and transmit their forcing.

### Muscle architecture

The design of the leg muscles was based on the Peano-HASEL actuator, which was chosen for its well-rounded performance among HASEL actuators[18]. We made several modifications to the actuator design that enhanced its force output profiles while reducing the actuators' weight; those changes optimized the muscles' specific energy.

To increase the force output, the width of the pouch was expanded from the standard Peano-HASEL design[40] from 60 mm to 90 mm. This expansion in width resulted in a directly proportional increase in force output. In addition, stacking multiple HASEL actuators in parallel achieved a cumulative effect, resulting in a linear increase in force output. To minimize the overall mass of the robot without compromising its force capabilities, the number of layers for each muscle pack was optimized based on the required force output profiles while reducing the leg inertia. This approach enabled us to achieve the necessary force levels for the robot's locomotion tasks. Moreover, individual connections were established between each muscle layer, the joint lever arms, and the tensioning mechanism, reducing strain loss from uneven load distribution across the muscle layers.

The original 20-mm-long HASEL pouches were shortened to 10 mm in length to reduce the muscle weight. This change led to a reduction in the weight of the filled liquid dielectric. Additionally, lightweight glass fiber plates of 0.5 mm thickness were used to replace the 5 mm thick acrylic boards typically used for connecting the actuator's plastic film and the tendon. Because of these modifications, the muscles achieved a specific energy of 7.4 J/kg (including the masses of tendon attachments and electrical connectors), enhancing the

original 2.0 J/kg. For the actuator's power output, the maximum specific power increased to 580 W/kg (Supplementary Fig. 7) from the original 160 W/kg[40].

### Self-sensing mechanism

We use a self-sensing system to measure the actuators' positions and corresponding joint angles without relying on encoders. For this, we leverage the capacitance change in HASEL muscles. During actuation, the capacitance of the HASEL actuator changes according to its displacement[53]. This relationship between capacitance and displacement allows us to determine displacement by measuring the capacitance. Specifically, larger capacitance values correspond to more zipping in the pouch, leading to larger displacements, while smaller capacitance values indicate smaller displacements. The joint angles were estimated by leveraging this mechanism, which involved monitoring the applied voltage $U(t)$ and current flow $I(t)$ from the high-voltage power supply to the muscles. The capacitance value $C(t)$ was computed from the current $I(t)$ and the voltage $U(t)$ as shown in the following formula:

$$Q(t) = \int_0^t I(t)\mathrm{d}t \tag{1}$$

$$C(t) = \frac{Q(t)}{U(t)} \tag{2}$$

### Transmission mechanism

We optimize the profile of the force output during locomotion using a nonlinear angle-dependent moment arm system instead of a linear transmission system. This approach aimed to tailor the angle-torque output profile to meet the required performance criteria for locomotion (Supplementary Fig. 2). Peano-HASEL artificial muscles exhibit a distinctive, highly nonlinear force-strain curve[40]. The high force range is susceptible to the backlash of the mechanical system, while the high strain range lacks sufficient force for effective actuation under loaded conditions. We address these challenges and enhance performance by implementing a nonlinear moment arm that flattens the torque profile. Not that we did not choose the more recent but less mature high-strain Peano-HASELs[58] as an alternative approach due to their less established fabrication technique.

### Experimental setups

Three types of setups (Supplementary Fig. 8) were employed in the ten experiments, as outlined in Supplementary Table 4. In the frame mount setup, the robot body was mounted on a stationary aluminum frame. In contrast, the 1D-boom featured a single degree of freedom (DOF) rotational joint perpendicular to the robot's sagittal plane, enabling vertical locomotion. The 2D-boom setup incorporated an additional rotational joint perpendicular to the robot's transverse plane, allowing for horizontal locomotion around the pivot center.

For all test platforms, the experimental system was comprised of the robot equipped with electrohydraulic muscles and encoders, high-voltage amplifiers, a standalone computer, and a data acquisition (DAQ) system (Fig. 2a). The electrodes of the electrohydraulic muscles were directly connected to the high-voltage amplifiers, which applied a consistent supply of a high voltage to the muscles while concurrently monitoring the applied voltage and current flow. Transmission of input signals from the standalone computer to the high-voltage amplifiers was accomplished via the DAQ. Furthermore, the DAQ system monitored voltage and current values from the high-voltage amplifiers and joint angle measurements from the encoders integrated within the robot's hip and knee joints.

The weight of the boom arm and cables from the amplifiers were compensated by a counterweight attached to the opposite side of the

robot on the boom arm. Additionally, the weight of the robot itself was varied in specific experiments. In experiments #5 and #10, the robot mass was set to 198 g, which differed from the original weight of 225 g. In experiment #8, the robot's mass was varied from 198 g to 391 g (−27 g to +166 g) to test the impact of weight on COT. During each experiment, the robot was driven by combinations of multiple high-voltage amplifiers, selected according to the required power output for each locomotion (Supplementary Table 5).

## Vertical jumping agility and jump frequency

Vertical jumping agility $V_{jump}$[43] and jump frequency $F_{jump}$[43] were calculated as follows:

$$V_{jump} = \frac{H_{jump}}{T_{jump}} = \frac{H_{jump}}{t_{stance} + t_{apogee}}, \tag{3}$$

$$F_{jump} = \frac{1}{T_{jump}}, \tag{4}$$

where $H_{jump}$ is the jump height, $t_{stance}$ is the total stance time from the onset of actuation, and $t_{apogee}$ is the flight time from when the jumper leaves the ground until the apogee of a jump (when the vertical velocity is zero).

## Statistical analysis of the gait range of motion

The leg's foot trajectory at each frequency was compared through the trajectory's enclosed area size. The mean and standard deviation were calculated from multiple motion cycles after reaching a steady limit cycle for a given frequency. Supplementary Table 6 provides the number of motion cycles used for the calculation at each frequency.

## Open-loop feed-forward controller

The open-loop feed-forward controller utilized a cyclic signal generator capable of generating ramped square waves and modified sine waves according to the following equation:

$$V_{in} = A \sin^2 \left( \frac{2\pi f t - \theta}{2} \right), \tag{5}$$

In Eq. 5, $A$ denotes the voltage amplitude, $f$ refers to the cyclic frequency, $t$ represents time, and $\theta$ represents the phase shift for each muscle pack. The parameters for the ramped square waves included voltage amplitude, duty cycle, ramp-up speed, and phase shift. Similarly, the parameters for the modified sine waves are the voltage amplitude, frequency, and phase shift. This controller was used for all the experiments in this study except for the trajectory tracking experiments.

## Cost of transport

The cost of transport was calculated as follows:

$$COT = \frac{E_{cons}}{mgl}, \tag{6}$$

where $E_{cons}$ represents the energy supplied from the high voltage power supply, $m$ is the robot mass, $g$ stands for gravitational acceleration, and $l$ signifies the distance the robot moves during the experiment's duration.

## DC-motor leg

The DC-motor leg was designed to have the same size and weight as the PELE leg. Two motor drive modules (DRV8302, Texas Instruments) control the two DC motors (T-Motor Antigravity MN4006). As for the PELE leg, coupling parts printed from PLA+ (eSun) and carbon fiber

rods were utilized for the leg's skeleton. To ensure weight equivalence with the PELE leg, the DC-motor leg was counterbalanced with additional weights, resulting in a weight of 224.4 g on the boom. Miniature rotary magnetic encoders (RM08, RLS) were installed on the joints for position feedback. The position control of the DC motors was implemented using the SimpleFOC library[59]. A current sensor (ACS724, Allegro MicroSystems) was used to monitor energy consumption.

## Energy and power consumption

The energy consumption $E_{cons}$ and average power consumption $P_{cons}$ during the squatting task were calculated by measuring the applied voltage and current flow from the high-voltage power supply to the leg using the following equation:

$$E_{cons} = \int_0^T \max(0, P(t)) dt = \int_0^T \max(0, U(t)I(t)) dt, \tag{7}$$

$$\bar{P}_{cons} = \frac{E_{cons}}{T}, \tag{8}$$

$$\bar{P}_{ideal} = \int_0^T \frac{\max(0, U(t)I(t)) dt}{T}, \tag{9}$$

where $P(t)$ represents the robot's power consumption, $U(t)$ denotes the applied voltage, and $I(t)$ describes the supplied current.

## Energy recuperation potential

The energy recuperation potential $E_{RP}$ was calculated as follows:

$$E_{RP} = \frac{E_{recup}}{E_{cons}} = \frac{\left| \int_0^T \min(0, P(t)) dt \right|}{\int_0^T \max(0, P(t)) dt} \tag{10}$$

where $E_{cons}$ represents the energy supplied from the high voltage power supply (the red-colored area in Fig. 5d), and $E_{recup}$ stands for recuperated energy to the power supply (the blue-colored area in Fig. 5d).

## Closed-loop feedback controller

The closed-loop feedback controller utilized a cascaded architecture incorporating a high-level task-space controller, a joint-space controller, and a low-level HASEL actuator controller. The control block diagram is provided in Supplementary Fig. 3a, demonstrating the transformation from the desired task space input to the final robotic leg response.

The control architecture was initially established at the actuator level, using an experimental platform equipped with a laser sensor to measure the position of a single muscle pack. This setup enabled the development of the HASEL actuator controller. Subsequently, a joint-space controller for a single joint was implemented based on the antagonistic pairing of two separately controllable muscle packs. This setup utilized encoder measurements together with the previously established HASEL actuator controller. Lastly, the leg setup with two joints was used to create the task-space controller, which operates on top of the lower-level joint space and HASEL actuator controllers. This integration allowed for effective task-space control of the robotic leg (Fig. 2c).

To ensure real-time system performance, the entire framework was implemented in C++, with multiple threads utilized to handle various tasks, including data acquisition (DAQ) communications, control loop, measurements/filtering, and data logging. The control loop was maintained at a fixed frequency of 500 Hz.

At the core of the control system, a proportional integral derivative (PID) controller was utilized as the low-level actuator controller. This controller received desired tendon positions derived from the leg's inverse kinematics and a mapping function that links joint angles

to tendon positions, along with actual tendon positions obtained from encoder measurements. Consequently, the PID controller generated the commanded voltages required for each muscle pack.

Supplementary Fig. 3b illustrates the time evolution plots of various signals, including the leg tip's Cartesian positions, joint and tendon positions, and control signals. In addition, Supplementary Fig. 3b shows the transformation from the high-level task-space planner to the low-level HASEL controller and highlights the control system's integration.

Supplementary Fig. 4 shows the root mean square (RMS) joint error as a function of the commanded joint position frequency. The results show that the control system can operate well at frequencies under 1 Hz and maintain the desired joint positions despite the increased frequency of joint commands. The results also highlight the trade-off between desired speed and accuracy, with higher frequencies leading to increased errors.

To quantify the joint errors, we employed the following equation:

$$E_i = \sqrt{\frac{1}{N}\sum_{j=1}^{N}(q_{i,m}(j) - q_{i,d}(j))^2} \text{ for } i \in \{\text{hip,knee}\}. \quad (11)$$

where $q_{i,m}$ is the actual position, $q_{i,d}$ is the desired position for each joint $i$, and $N$ is the number of measurements. The overall RMS error is calculated as the average of the RMS errors for the hip and knee joints.

## Data availability

The paper and/or the supplementary information contain all the data needed to evaluate the conclusions. Correspondence and material requests should be addressed to Robert K. Katzschmann and Christoph Keplinger.

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

## Acknowledgements

We thank An Mo, Bernadett Kiss, Emre Cemal Goenen, and Alexander Badri-Spröwitz from the Dynamic Locomotion group of MPI-IS for their advice on the experimental setup and data analysis. We thank Oriol Llorà and Semi Kim for assisting with fabricating Peano-HASEL artificial muscles. We further thank Bradley Nelson for lending his group's high-voltage amplifier. This work was in part made possible by support from the following sources: Credit Suisse's donation to the ETH Foundation to create a new chair for robotics at ETH Zurich (R.K.K.); Max Planck Society (C.K.); Zurich Heart project #2022-FS-320 (T.J.K.B. and S.-D.G.); Swiss National Science Foundation Grant #200021_215489 (T.J.K.B., A.K., and S.-D.G.); National Centre of Competence in Research Robotics Grant (A.K.); Swiss Government Excellence Scholarship for PhD studies (A.K.); ETH Grant #ETH-17 22-1 (A.K. and T.J.K.B.); RobotX Research program #RX-03-22 (T.J.K.B.); International Max Planck Research School for Intelligent Systems (IMPRS-IS) (T.F.); Max Planck ETH Center for Learning Systems (T.J.K.B., T.F., R.K.K., and C.K).

## Author contributions

Conceptualization, funding acquisition, project administration, and supervision: C.K. and R.K. Methodology: T.J.K.B., T.F., A.K., P.A., P.R., M.P., S.-D.G., and R.K. Investigation: T.J.K.B., T.F., A.K., P.A., P.R., M.P., J.W., S.-D.G., Y.Z., X.W., and R.K. Visualization: T.J.K.B., T.F., A.K., S.-D.G., and R.K. Writing—original draft: T.J.K.B., T.F., A.K., S.-D.G., S.L.Z., and R.K. Writing—review and editing: T.J.K.B., T.F., C.K., and R.K.

## Competing interests

C.K. is a co-inventor on three patents covering the fundamentals and basic designs of HASEL actuators (The assignee of all three patents is "The Regents of the University of Colorado": US Patent 10995779B2, granted 2021-05-04; US Patent 11486421B2, granted 2022-11-01; US Patent 11408452B2, granted 2022-08-09). CK co-founded Artimus Robotics, a start-up company that commercializes HASEL actuators. The remaining authors declare no competing interests.
