## [Peer Review File · Nature Communications]

REVIEWER COMMENTS

Reviewer #1 (Remarks to the Author):

The authors report a tethered robotic leg, of mass approx. 200 grams, attached to a central pivot. The leg moves in a circular path by a hopping motion. The leg has two active joints, each driven in an antagonistic manner by two peano-HASEL actuators.

By controlling the 4 actuators, the robot “foot” can be displaced at up to 10 Hz. The authors study mostly the hopping gait, showing successful open-loop motion on several types of surfaces. The authors estimate cost of Transport and power usage. Some limited self-sensing is shown by measuring the capacitance of the compliant electrostatic actuator.

In my opinion, this work is not well suited for Nature Communications. The hasel actuators are well known. The claims, especially in the abstract, are vastly exaggerated. The paper should be rewritten with toned down language, and submitted to a more specialized journal.

The papers makes a number of claims that are far too easily misunderstood, that take away from the underlying good work.

The title and abstract are misleading: The authors report a single leg, not a robot! Why not use the word “leg” in the title?

The title is excessive and unjustified: no new technology is reported that enables motor free robots. The original peano-hasel paper from Keplinger’s group could also have claimed as a title “no more motor”. As could many papers using soft actuators to make moving robots! What is special here to justify such a title?

The authors report “ a legged system adaptively traverses varying terrain”, but this is a misleading way of implying the robot can actually move itself: it cannot. It needs to be attached to a fixed pivot or the leg simply falls over. No self-balancing, no untethered operation.

The author claims “autonomous” but leg is attached to fixed pivot, and wired to four large external power supply!

The “leg jumps 40% of body height” ... but there is no body, since there is no robot. Only a leg is reported! Please, it is irritating for this reviewer to have to correct such blatant exaggeration. Dear

senior authors, please consider how publishing this work might affect your hard-earned reputations.

I don't follow the claims of "Inherent adaptivity". I can drive a car through mud, over gravel, over roots and on pavement. Does that show "Inherent adaptivity"? in the same way, I don't see why this same leg, if one replaced the peano-hasels with EM motors, would have any difficulty going over the grass, pebbles, or sand. I do not see why an EM-driven leg would need any complex closed-loop control, just as this peano-hasel driven leg does not need closed loop control

I have serious concerns about the claim that the peano-hasel driven leg is over 150x more power efficient than an electromagnetically driven leg. How were the EM motors chosen? Why do they need 100 W peak power? Even if I assume only 5% efficiency for the EM option, I can't see how the leg presented here is 100x more efficient.

The leg is driven by 4 multi-kg power supplies. Is this reflected in the analysis?

Do I understand correctly the energy computation was done for a robot that was not built, but that the authors think might be built with better materials, at some point in the future (ie subtracting 27 g from the real mass)? I do not consider this to be a valid method. Also, the mass of the assumed power supply is orders of magnitudes lower than the mass of the real power supplies listed in Extended Data Table 3. This cavalier approach to computing efficiency and power are of concern.

Reviewer #2 (Remarks to the Author):

Overall this is an interesting paper demonstrating some important advances in soft actuator design and implementation. The following comments are relatively minor but should be addressed to make the work more complete.

With reference to the energy efficiency of the squatting posture (lines 9 through 97 etc.). Is this a fair comparison? Could the electromagnetic motor system be designed such that uses minimal energy in the squatting experiment?

Unfortunately, the video for agile locomotion is corrupted and could not be played.

Figure 2 Why are the units shown in this range (-11 to -7) instead of from a 0,0 origin?

Line 143. How was optimization carried out?

Line 145 What are the actual values of power and weight for these actuators?

Figure 3B The time axis should be labeled appropriately with units

Figure 3B How was the timing of the flexor and extensor input signals determined? Clearly the start of each smooth sinusoidal flexor muscle activity partially overlaps with the pulsatile extensor activation, is this to provide stability through an increase in joint stiffness? I presume that the flexor activity is also used to bring the leg back to its starting stance position faster than gravitational acceleration.

It would be interesting to characterize the work loop properties of the different muscles for comparison with known work loops in jumping animal legs.

Figure 4, panel a needs actual time units indicated

The stiffness of the muscle and their attachment points should be stated, it would be helpful to have passive and active work loop information (i.e. stress-strain relationships) of the actuator itself and in combination with jointed arm.

It would be a good idea to have a more recent reference for the energetic cost of animal locomotion, the one used here was from 1975. This should also be confined to terrestrial legged locomotion.

Eg Halsey, L.G., White, C.R. Terrestrial locomotion energy costs vary considerably between species: no evidence that this is explained by rate of leg force production or ecology. *Sci Rep* 9, 656 (2019). <https://doi.org/10.1038/s41598-018-36565-z>

Pontzer H. A unified theory for the energy cost of legged locomotion. *Biol Lett.* 2016 Feb;12(2):20150935. doi: 10.1098/rsbl.2015.0935. PMID: 26911339; PMCID: PMC4780550.

I am unclear about the cost of transport estimates. Although the total cost and net cost for other robots are included in figure 5, only the net cost for this robot are shown. The net cost of transport is calculated from the electrical energy supplied from the power supply to the leg. You should include an estimate of power lost in the high-voltage conversion because this will affect the energy storage capability and untethered lifetime of the robot.

How was the crawling gait to defined? There is no definition or demonstration of crawling.

The energy efficiency of maintained squatting posture is an exciting finding and the simple open-loop obstacle detection system is also an appealing feature of this robot design. Of course it is currently limited to a simple mode switching control feature but I'm sure this could be expanded to include additional responses or gates dependent on the proprioceptive feedback.

It is pleasing to see some of the limitations of this creative design honestly discussed. Although it's not necessary to include here, the other limitation for practical applications is obviously the high voltages necessary for the HASEL actuators. I suspect that this danger could be mitigated by appropriate packaging.

Reviewer #4 (Remarks to the Author):

General Comments:

This manuscript describes the fabrication and testing of a bio-inspired leg composed of Peano-HASEL electrohydraulic actuators. The leg, attached to a 2D-boom, is shown to passively adapt its gait while using open-loop control to traverse gravel, grass, sand, small stones, and larger rocks. Frequency-dependent jumping and cyclic motions are analyzed. The leg is capable of dynamically adapting its gait in response to obstacles using inherent self-sensing capabilities. Comparisons are made to a fabricated conventional DC-motor-driven leg along with cost of transport comparisons with other published legged robots. The manuscript's leg shows promising results and could offer an energy-efficient, highly adaptive, and self-sensing alternative to conventional robotic legs.

Details:

1. The PELE is attached to a boom for locomotion and jump tests. In Fig. 5b, the PELE is compared to multiple-legged robots and land animals which do not use a boom for support and are instead self-standing. This makes the COT comparison feel a little incomplete. At the least, this discrepancy should be noted in the main text of the manuscript. Additionally, the use of the boom support should be mentioned in the abstract for clarity.

2. Please mention either in the Fig. 5 caption or in the main text whether the COT for PELE is a net or total COT measurement.

3. Please include a discussion of the expected cycles to failure as well as the expected durability of the actuators (Durability in terms of reactions to external contact and punctures).

4. Please provide an explanation or hypothesis as to why the gait range of motion increases and then decreases as the gait frequency increases in Fig. 3e.

5. The HASEL acronym should be first spelled out on line 69 rather than line 108.

6. Comparisons with PELE were done with a DC-motor-driven leg of comparable size and weight. Care should be taken especially during the energy efficiency comparisons as the DC-motor leg likely can support a higher mass payload than PELE, such as shown in the DC-motor systems shown in Fig. 5b. For a more apt comparison the DC-motor leg should have been undersized in terms of size and/or weight and should instead be sized to match the mass payload capacity of the PELE. Alternatively, if the DC motor were controlled via the drive modules in a way such as to match this payload capacity that would also be acceptable. If that is the case in the current manuscript it is not clear.

7. Continuing from 6., when stating that PELE required ~0.6% of the energy of a comparable DC-motor-driven leg please instead compare it with a DC-motor-driven leg that either matches the total mass payload capacity of the PELE or show/state that only energy as needed is supplied to the DC motors until it matches the payload capacity of PELE, without excess.

8. The claim that all electromagnetic leg designs without closed-loop control would fail to traverse grass, sand, gravel, pebbles, and large rocks is quite bold. This claim needs to be substantiated more in the text.

**Response to referees of the manuscript:
“Electrohydraulic musculoskeletal robotic leg for agile, adaptive, and energy-efficient locomotion” (NCOMMS-23- 49570)**

Dear Referees,

We thank you for handling our manuscript NCOMMS-23-49570 and for providing your very constructive comments. Based on your remarks, we conducted a comprehensive set of new experiments and addressed your suggestions, questions, and concerns throughout the manuscript; our changes and additions are detailed in the following pages, where we address your comments point by point. Specifically, we would like to highlight the following **major additions and changes to our manuscript**:

1. **Test platform:**

We have now adapted the manuscript and refer to the system as a robotic leg on a boom arm.

2. **Comparisons of power consumption and cost of transport (COT):**

We have now explained in the manuscript the underlying mechanism of the low power consumption of PELE (zero power consumption while holding a posture), the reasoning behind the DC-motor leg design (equal simplicity and versatility), and specific numbers of power consumption (actual measurement and expected values using energy recuperation technology)

3. **Exaggerated language:**

We have adapted the language used in our manuscript, changed the title, and toned down the abstract.

4. **Discussion on durability:**

We have added a discussion on the durability of the system.

5. **Additional experiments:**

We have analysed the work loop of the artificial muscles and characterized the adaptive stiffness of the system and added these results to the manuscript.

Table of contents for the responses:

Response to Reviewer #1 page 2ff

Response to Reviewer #2 page 13ff

Response to Reviewer #4 page 25ff

Color coding for the individual responses:

Reviewer's remark

Our response

Changes made to the text.

Response to Reviewer #1

Remark 1-1:

The claims, especially in the abstract, are vastly exaggerated. The paper should be rewritten with toned down language, and submitted to a more specialized journal. The papers makes a number of claims that are far too easily misunderstood, that take away from the underlying good work.

We would like to thank the reviewer for taking the time to provide us with detailed feedback, which helped us to improve our work further. We made substantial changes throughout the manuscript to adapt and tone down the claims, avoid misunderstandings, and improve the clarity of descriptions of our scientific findings. We feel that these changes and the new experiments added as part of this revision (as described below) have made our paper suitable and appealing to a broad audience.

Remark 1-2:

The title and abstract are misleading: The authors report a single leg, not a robot! Why not use the word “leg” in the title?

Following the reviewer’s suggestion, we adapted the title, abstract, and manuscript and now refer to our system as a robotic leg instead of a robot.

We adapted the title to:

~~No more motors — electrohydraulic~~ Electrohydraulic musculoskeletal ~~robots~~ robotic leg for agile and adaptive, yet energy-efficient locomotion.

In the abstract, we now define our system more clearly and use the term ‘leg’.

Line 24ff:

We propose a bio-inspired musculoskeletal leg architecture driven by antagonistic pairs of electrohydraulic artificial muscles. ~~Our legged system~~ Our leg is mounted on a boom arm and can adaptively ~~traverses~~ hop on varying terrain in an energy-efficient yet agile manner ~~and detects~~. It can also detect obstacles through capacitive self-sensing. The leg performs powerful and agile gait motions beyond ~~5 Hz~~ 5 Hz and high jumps ~~of up to 40% body~~ % of the leg height.

Remark 1-3:

The title is excessive and unjustified: no new technology is reported that enables motor free robots. The original peano-hasel paper from Keplinger’s group could also have claimed as a title “no more motor”. As could many papers using soft actuators to make moving robots! What is special here to justify such a title?

We agree with the reviewer and, in retrospect, regret that we went with a title designed to be catchy and easy to remember. The new title is toned down and focuses on our key findings:

~~No more motors — electrohydraulic~~ Electrohydraulic musculoskeletal ~~robots~~ robotic leg for agile and adaptive, yet energy-efficient locomotion.

Remark 1-4:

The authors report “ a legged system adaptively traverses varying terrain”, but this is a misleading way of implying the robot can actually move itself: it cannot. It needs to be attached to a fixed pivot or the leg simply falls over. No self- balancing, no untethered operation.

The author claims “autonomous” but leg is attached to fixed pivot, and wired to four large external power supply!

We thank the reviewer for the remark and the opportunity to provide a more balanced explanation of our work on an electrohydraulic leg. We adapted the manuscript to clarify the use of a boom arm and the tethered operation of the leg. At the same time, we now highlight that our work is building on an essential line of research that identified the critical component of legged locomotion as the leg itself [Badri-Spröwitz et al., 2022; Ruppert et al., 2019]. That line of research studies different driving mechanisms and geometries for legs attached to boom arms to study fundamental aspects of legged locomotion, including controls, energy efficiency, and actuation.

References:

Badri-Spröwitz et al., Science Robotics, 2022,

<https://doi.org/10.1126/scirobotics.abg4055>

Ruppert et al. Frontiers in Neurorobotics, 2019,

<https://doi.org/10.3389/fnbot.2019.00064>

Changes made to the abstract:

Line 25ff:

~~Our legged system~~Our leg is mounted on a boom arm and can adaptively traverseshop on varying terrain in an energy-efficient yet agile manner ~~and detects. It can also detect~~ obstacles through capacitive self-sensing. The leg performs powerful and agile gait motions beyond ~~5Hz~~5 Hz and high jumps ~~of up to 40% body % of the leg height. While an electromagnetic leg design without closed-loop control would fail, our~~Our leg's tuneable~~tunable~~ stiffness and inherent adaptability allow it to ~~traverse~~hop over grass, sand, gravel, pebbles, and large rocks using only open-loop force control.

In the abstract, we used the word ‘autonomous’ to describe legged robots in general, not specifically referring to PELE: “*Autonomous locomotion in unstructured terrain demands an efficient, agile, and adaptive architecture.*” However, the reviewer's feedback prompted us to recognize that the scope of locomotion is not exclusively autonomous. Therefore, we have revised the sentence accordingly:

Change made to the abstract:

Line 20f:

~~Autonomous~~Robotic locomotion in unstructured terrain demands an ~~efficient, agile, and adaptive, and energy-efficient~~ architecture.

Changes made in the main manuscript:

Line 99f:

The leg can ~~freely traverse~~hop on varying terrain using ~~purely~~only its inherent proprioception ~~without the need for closed-loop control, relying on knowledge of the joint angle (for example, encoder values).~~

Remark 1-5:

The “leg jumps 40% of body height” ... but there is no body, since there is no robot. Only a leg is reported! Please, it is irritating for this reviewer to have to correct such blatant exaggeration. Dear senior authors, please consider how publishing this work might affect your hard-earned reputations.

Following the reviewer’s suggestion in Remark 1-2, we adapted the title, abstract, and manuscript to consistently refer to our system only as a robotic leg (on a boom arm) rather than a robot. The term “body height” was changed to “leg height.”

Changes made in the abstract:

Line 24ff:

We propose a bio-inspired musculoskeletal leg architecture driven by antagonistic pairs of electrohydraulic artificial muscles. ~~Our legged system~~Our leg is mounted on a boom arm and can adaptively ~~traverseshop on~~traverse varying terrain in an energy-efficient yet agile manner ~~and detects~~. It can also detect obstacles through capacitive self-sensing. The leg performs powerful and agile gait motions beyond ~~5 Hz~~5 Hz and high jumps ~~of up to 40% body % of the~~up to 40% of the leg height.

Changes made in the main manuscript:

Line 71f:

We use the legs isolated from complete robotic systems because the leg is a crucial component in legged robotics to study the impact of actuator choice (36,37).

Line 89f:

PELE also performed powerful and agile gait motions beyond 5 Hz, linear motions beyond 10 Hz, and high jumps at over 40 % of ~~its body~~the leg height. ~~The musculoskeletal design of (Fig. 1b top left).~~

Line 369f:

PELE (iv) showed agile gait motions at over 5 Hz ~~and allowed for~~, allowing rapid jumps of more than 40 % of the ~~robot’s body~~leg height.

Remark 1-6:

I don’t follow the claims of “Inherent adaptivity”. I can drive a car through mud, over gravel, over roots and on pavement. Does that show “Inherent adaptivity”? in the same way, I don’t see why this same leg, if one replaced the peano-hasels with EM motors, would have any difficulty going over the grass, pebbles, or sand. I do not see why an EM-driven leg would need any complex closed-loop control, just as this peano-hasel diven leg does not need closed loop control.

We thank the reviewer for the remarks on the inherent adaptivity of PELE, which gave us an opportunity to make our argument more straightforward.

Wheeled systems are very good at operating on flat terrain, yet more complex terrain becomes difficult to handle; steep climbing, for example, would not be possible. Wheeled-type locomotion is usually resilient to wheel slippage because of its infinite range of rotational motion, whereas legged locomotion requires adaptivity for

locomotion under disturbance due to its end-to-end motion. This adaptivity is either gained through an inherently adaptive actuator/structure or added to the system via hardware or software. This makes wheeled and legged locomotion inherently different.

Specifically, we claim PELE is inherently adaptive because PELE is controlled by feed-forward “force control” without a joint angle encoder; the voltage applied to HASEL actuators is directly related to the force output of the actuator. In contrast, a low-level (joint space) controller of a DC-motor-driven leg requires a position feedback loop using a joint angle encoder in addition to torque control based on the motor current.

The control of PELE is equivalent to a situation where a DC-motor-driven leg would be given only a cyclic reference q-current (current used for torque generation) signal on a motor (see Figure R1) while continuously trying to locomote in varying terrain. Without knowing the exact motor position, the locomotion will be corrupted by noise caused by environmental interactions (see formulas in Figure R2). To avoid this type of position corruption, a motor would need a parallel spring and end dampers to define a “natural position” of the joint (hardware solution) between two states of maximum joint rotation, which leads to further complexity and torque reduction of the joint. In contrast, the adaptivity of PELE stems from the use of antagonistic pairs of Peano-HASEL actuators, which inherently create a minimum energy resting position by acting as parallel springs and end dampers. In the same way, a biological joint becomes adaptive from the specific force-strain characteristics of natural muscles and their antagonistic placement around the joint.

Typical leg controller

Figure R1: Typical leg controllers (<https://doi.org/10.1109/ICRA.2018.8460731>) include a joint position controller (blue triangle in top left schematic and break out view below)

(<https://docs.odriverobotics.com/v/0.5.4/control.html>); in contrast, robotic legs driven by Peano-HASELs do not require a joint position controller.

- Wheeled locomotion (car, train, drone, etc.)
 - Rotational motion → infinite motion → only torque control and speed sensor is needed
- Legged locomotion
 - Reciprocating motion → end-to-end motion → position control is needed
 - **Motion can be unstable** when only with torque control

- Natural position cannot be defined
- Requires **parallel spring** and **end dampers** to define the natural position

$$\theta(t) = \iint \ddot{\theta}(t) dt$$

$$\theta(t) = \iint \frac{\tau(t)}{I} dt^2$$

$$\theta(t) = \iint \frac{\tau(t) + \text{noise}(t)}{I} dt^2$$

t: time
 θ : joint angle
 τ : joint torque
 I: moment of inertia

In antagonistic pairs of HASELs

- Antagonistic pairs of muscles → natural spring
- Reduced force at high strain → natural end damper

Figure R2: Fundamental differences between wheeled and legged locomotion, as well as legged locomotion based on electromagnetic motors and legged locomotion based on antagonistic pairs of Peano-HASEL muscles.

Based on the discussion above, we adapted the manuscript as follows:

To better clarify our argument on adaptivity, we changed all indications of open-loop control regarding our leg to open-loop force control and removed claims about the DC-motor driven system:

Line 28ff:

Our leg's tuneable stiffness and inherent adaptability allow it to traverse hop over grass, sand, gravel, pebbles, and large rocks using only open-loop force control.

We also added lines 55ff:

Although ongoing research aims to improve these limitations (34), electromagnetic motor-driven systems require complex controls to “approximate compliance” through software — the control loop speed limits software-based compliance, resulting in reduced safety during accidental fast impact. Additionally Direct driving transmissions provide back-drivability to the system, on the other hand, the actuators continuously consume energy while holding a payload unless they are fitted with complex clutch systems (35).

Line 87ff:

Requiring only open-loop control, the leg Moreover, the musculoskeletal design and the electrohydraulic actuators of PELE enabled an open-loop force controller via only regulating applied voltage to manipulate and locomote the leg.

Line 122ff:

In contrast to electromagnetic motors, where supplied current correlates to actuator torque output, voltage applied to an electrohydraulic actuator correlates to actuator force. The controlled actuator force output, combined with the antagonistic pair of muscles and the actuator's force-strain characteristics, allows the leg to locomote in open-loop force control mode without requiring a joint angle encoder. PELE is, therefore, inherently adaptive.

Remark 1-7:

I have serious concerns about the claim that the peano-hasel driven leg is over 150x more power efficient than an electromagnetically driven leg. How were the EM motors chosen? Why do they need 100 W peak power? Even if I assume only 5% efficiency for the EM option, I can't see how the leg presented here is 100x more efficient. The leg is driven by 4 multi-kg power supplies. Is this reflected in the analysis?

We thank the reviewer for voicing their concern and for giving us the opportunity to explain our results in more detail. We measured the difference in power consumption for our electrostatically driven leg compared to our electromagnetic motor-driven leg in the specific task of continuous squatting and found it to be 83 times smaller for the electrostatic version. In the case when using energy recuperation, that difference would increase even further.

To compare the legs fairly, we implemented specific design constraints. We wanted the total system weight to be similar for electrostatic and electromagnetic legs. The electromagnetic motor was implemented in direct drive mode (without a geared transmission) to achieve similar back-drivability, bandwidth, simplicity, and inherent impedance of the leg's joints. Importantly, both the DC-motor leg and PELE were designed to perform a variety of tasks (hopping, rapid gait motion, squatting, precise controllability), and neither leg was specifically optimized for squatting.

We choose the task of squatting with the intermediate position holding to highlight fundamental differences in the energy efficiency of different driving technologies. Due to the nature of a direct drive motor setup and electromagnetic motors, the electromagnetic leg has significant power consumption, especially when statically holding a deep squat position. When pushing off from the deep squat position, an even higher amount of power (100 W) is needed. Compared to that, the power consumption for pairs of electrostatic actuators when statically holding a position is negligibly small (see Table R1 on the new Extended Data Table 1) – electrostatic systems are an excellent option for operations with extended holding force requirements. So, depending on the amount of static holding requirements within a specific task, the difference in efficiency can become arbitrarily large [Krimsky et al., 2024].

We added subfigure 5f to the manuscript (see below **Error! Reference source not found.** in this response letter) regarding steady-state torque requirements for our leg

systems. The black line is the required quasi-static torque for the different squatting positions. The blue line is the HASEL torque output profile, and the orange line is the required motor output. In addition to the requirement of having an equal system weight, the motor was also chosen so that its torque output is within the output range of the HASEL leg system.

Following widely used research conventions in the field of legged locomotion, we set out to study the intrinsic energy efficiency of legged robotic systems – therefore, the power supplies of PELE or the EM-motor-driven leg are not part of the calculation. To give a rough estimation of the weight of the power supply needed to drive PELE for the squatting task, a 65g compact high-voltage power supply [Schlatter et al., 2018] would be sufficient.

Mentioned References:

Krimsky, et al, 2024, *Science Robotics*, <https://doi.org/10.1126/scirobotics.adj7246>

Schlatter et al. 2018, *HardwareX*, <https://doi.org/10.1016/j.ohx.2018.e00039>

Table R1: **Extended Data Table 1 | Motor – HASEL actuator equivalence.** While holding a constant position, a motor-driven system In HASEL actuators, power consumption during holding a posture does not relate to exerting torque.

Metrics	DC-motor	HASEL
Output	Torque	Force
Control	Current	Voltage
Sensor	Series resistor	Parallel resistor
Power consumption during holding position	$P = VI = I^2R \propto \tau^2$	$P = VI = \frac{Q}{c} \dot{Q} \sim 0$

We also added Fig. 5f, and updated Fig. 5e, where we now clearly distinguish between cases when using energy recuperation technology and cases without energy recuperation:

Figure R3: **Fig. 5 | Energy-efficient locomotion.** **a**, The leg hopped forward using an energy-efficient gait. **b**, The PELE's minimum net cost of transport (COT) of the leg compared to other legged robots. The different weights were simulated by attaching systems. PELE performed a hopping (triangle) and crawling (circle) locomotion. We detached and attached counterweights to the boom arm to investigate the effect of weight change. As a general reference, the 100 % COT line indicates the metabolic cost of land animals (50). Total COT includes all the power consumption on the robot (not including computers, drivers, cooling fans, etc.). Net COT accounts only for the power consumption of actuators. The PELE's mass of PELE (225 g) was varied from 198-27 g to 394+166 g. The hopping mode was most efficient for the lower weight weights, and the crawling mode was most efficient for the larger weights (COT in Extended Data Table 5.2). **c**, Thermal images of PELE and a DC-motor-driven leg during squatting motion. PELE was compared with a DC-motor leg of comparable size and weight and torque output. The image shows no observable thermal energy loss for the HASEL actuator. **d**, Comparison of power consumption during squatting. PELE showed the potential for energy recovery from the gravitational potential energy. **e**, Comparison of the consumed energy over time. For PELE, an area is shown. The lower line indicates the energy consumption under perfect recuperation, and the upper line indicates no recuperation. **f**, Analysis of the quasi-static torque needed to hold the leg in different

positions (knee angles). Orange lines indicate the DC motor's torque and power consumption. The blue line indicates the torque for PELE's actuators at different knee angles.

We made changes to the abstract (now giving number without the use of energy recuperation technology):

Line 30ff:

The electrohydraulic leg features a low cost of transport (~~0.69 to 0.83~~0.73), and while squatting, it consumes only a fraction of the energy (~~0.6~~1.2 %) compared to its conventional electromagnetic counterpart.

Line 111ff:

Each electrohydraulic muscle is a Peano-HASEL (~~hydraulically amplified self-healing electrostatic actuator~~), ~~which is made up of~~comprising serially stacked actuator pouches (40)~~(39)~~. A single pouch is a polymer shell filled with liquid dielectric and covered with electrodes on either side. When different electric potentials are applied on the two electrodes of the pouch, charges are moved to the electrode, and electrostatic forces cause a shape change ~~on~~in the ~~actuator~~pouches. This shape change leads to a linear contraction Δx along the serially stacked actuator pouches. The contraction is reversed once the electrodes are discharged. Electrohydraulic actuators usually have catch states, where no additional energy is required to hold a position, even under load (41,42), except for a very small amount of charge leakage through dielectric layers that must be compensated. Therefore, PELE has very little power consumption while holding a posture, even when exerting a substantial joint torque (Extended Data Table 1).

In contrast to electromagnetic motors, where supplied current correlates to actuator torque output, voltage applied to an electrohydraulic actuator correlates to actuator force. The controlled actuator force output, combined with the antagonistic pair of muscles and the actuator's force-strain characteristics, allows the leg to locomote in open-loop force control mode without requiring a joint angle encoder. PELE is, therefore, inherently adaptive.

Line 265ff:

Notably, the leg achieved a minimum COT of ~~0.69 at a reduced weight of 198 g, and a net~~ COT of 0.73 at the original weight of 225 g, ~~and a net COT of 0.69 and 0.83 at a reduced weight of 198 g and an increased weight of 391 g, respectively.~~ Intriguingly, as the weight exceeded 225 g, the locomotion type that yielded the minimum net COT transitioned from hopping to crawling. Compared to other legged ~~robots, the~~ leg systems (Extended Data Table 3), PELE exhibited remarkably favourable COT values within this range of robotic mass (Fig. 5b, Extended Data Table 2). The net COT calculation does, however, not include the DC-DC high voltage conversion efficiency in the driving electronics of typically around 75 % (51,52).

Line 283ff:

In contrast to the DC-motor leg's average power consumption P_{cons} (Equation 8) of 25.3 W during the squatting task, PELE consumed 306 mW, which was 1.2 % of the DC-motor leg. Additionally, power consumption data indicated that PELE consumed

only 9.4 mW to maintain its elevated position, while the DC-motor leg consumed 17 W.

Please note that we used a direct-drive architecture for the benchmark DC-motor leg. State-of-the-art DC-motor-driven legs with higher transmission ratios are likely more efficient than a direct-drive DC-motor leg, ~~but are also more limited in back drivability, a key feature of PELE presented in this work.~~ and the DC-motor leg could be optimized for the specific squatting task. Nevertheless, our selection of the direct-drive approach was aimed at matching mass, back-drivability, bandwidth, and simplicity of the actuators used to drive PELE. This design choice results in versatility of the DC-motor leg to be used in different tasks, mirroring PELE where a single design achieved all the different tasks. For the DC-motor (T-MOTOR, MN4006), the selection criteria were its weight and its output torque; its weight was comparable to that of the Peano-HASEL stacks and its output torque met the quasi-static joint torque requirements (as illustrated in Fig. 5f). Notably, the power consumption of the DC-motor escalates with the square of the output torque (orange lines in Fig. 5f). By contrast, the Peano-HASEL exhibits constant, minimal power consumption across all torque levels while holding a posture (blue line in Fig. 5f). When PELE returned to its original posture, a major part of the potential energy gained from the leg standing up was sent back to the power supply (Fig. 5d). In the future, there is the potential to recuperate this energy coming back from the actuators through suitable driving electronics that yet must be designed. Assuming we can capture all the energy coming back from discharging the Peano-HASEL actuators, the average power consumption could reach a minimum of 43 mW (P_{ideal} , Equation 9) instead of the 306 mW reported here. We calculated the energy recuperation potential (E_{RP} , Equation 6) ~~10) of PELE~~ to be 85 %.

Line 390ff:

(ii) So far, we have only shown a single 2-DOF leg mounted on a boom, which limits the leg's locomotion to a circular path. Hexapedal, quadrupedal, and bipedal systems must be investigated to fully characterize the potential of this design architecture.
(iii) We only compared the energy efficiency for squatting. It would be important to extend the investigation into other types of motion, for example, continuously fast actuation like used for running.

Remark 1-8:

Do I understand correctly the energy computation was done for a robot that was not built, but that the authors think might be built with better materials, at some point in the future (i.e., subtracting 27 g from the real mass)? I do not consider this to be a valid method. Also, the mass of the assumed power supply is orders of magnitudes lower than the mass of the real power supplies listed in Extended Data Table 3. This cavalier approach to computing efficiency and power are of concern.

We thank the reviewer for the remark, which allows us to clarify our approach further and improve our explanations. The calculation for the energy consumption during squatting (discussed above) and the calculation of the cost of transport were done with the original mass of the leg and reported in the manuscript accordingly (excluding the boom arm, power supply, and computer controller). We further investigated the effect of varying leg masses (from -27g to +166g) on the leg's cost of transport. By reducing

and increasing the counterweight on the boom arm, we provide the reader with insights into the severity of the effect of weight changes on future developments on robotic legs. Future weight decreases could stem from improvements in actuator materials, and weight increases could stem from onboarding the driving electronics.

Regarding the weight of the power supply, please find our detailed rationale below. The HASEL leg was driven in the COT experiment by a single, stationary, one-channel power amplifier at the maximum of 6kV at 5mA shared between the hip and knee flexor. We changed the weights according to estimations based on the performance of an existing compact high-voltage power supply (1W, 1ch at 65g) [Schlatter et al., 2018]. Considering swapping the key power supply components (DC-DC converter and optocouplers) for our hopping requirements (ULTRAVOLT 6AA24-P30, HVP OPC10M), the mass of 172g (30W, 1ch) of a portable high voltage power supply would satisfy the requirements of the COT experiments and adding a battery (24V, 380mAh, 62g) would increase the mass to a total of 234g. We think this mass estimate is rather conservative since these power supplies were not optimized for weight. For example, recent research on HALVE actuators, a composite HASEL actuator using high permittivity dielectric films and run around 1 kV, has shown that power supplies using MOSFETs instead of optocouplers can achieve substantially lower weights [Gravert et al., 2024].

Cited References:

Schlatter et al. 2018, HardwareX, <https://doi.org/10.1016/j.ohx.2018.e00039>

Gravert et al., 2024, Science Advances, <https://doi.org/10.1126/sciadv.adi9319>

Changes in the main manuscript line 241ff:

We investigated the influence of the leg's mass on its energy efficiency. Therefore, we considered two scenarios: first, the potential reduction of the robot's mass achieved by further lightweighting of the actuators as detailed in Kellaris et al. (46)~~improved lightweight components,~~ and second, the increase of mass when adding mobile power supplies and batteries for untethered operation in the future (47–49)~~operations.~~ To test these scenarios, we adjusted the counterweight mass on the boom arm to achieve emulate the robot's mass in a range from 198 g to 391 g (-27 g to +166 g).

Response to Reviewer #2

Remark 2-1:

With reference to the energy efficiency of the squatting posture (lines 9 through 97 etc.). Is this a fair comparison? Could the electromagnetic motor system be designed such that uses minimal energy in the squatting experiment?

We thank the reviewer for the concern and for giving us an opportunity to explain our results in more detail. Regarding the question of fair comparison, please refer to our answer to Reviewer #1, Remark 1-7, for a detailed reply on energy efficiency. To make a DC-motor leg use minimal energy in squatting, we would have to add a clutch or a fairly high gearing ratio into the design, which would increase system weight and complexity [Krimsky, et al, 2024], while reducing the adaptability and compliance of the system.

Krimsky, et al, 2024, *Science Robotics*, <https://doi.org/10.1126/scirobotics.adj7246>

Remark 2-2:

Unfortunately, the video for agile locomotion is corrupted and could not be played.

We apologize for this mishap and have reuploaded the video with a different encoding that we hope is more widely compatible with most players. Please let us know if you have any issues.

Remark 2-3:

Figure 2 Why are the units shown in this range (-11 to -7) instead of from a 0,0 origin?

We thank the reviewer for the remark. We updated Fig. 2a in the manuscript (see Figure R4 in this response letter) to clearly show the origin of the coordinate frame used in this work – the origin is in the hip joint. To be consistent, we use the same coordinate frame when describing the tip trajectories.

We added the origin to Fig. 2a and adapted the subfigure's caption adding a reference also in the caption of Fig. 2c (see Figure R4 below).

Figure R4: **Fig. 2 | Electrohydraulic artificial musculoskeletal leg system.** **a**, System schematic of our leg. A computer interfacing with four high-voltage amplifiers, one for each muscle pack, i.e., hip flexor and extensor as well as knee flexor and extensor. Each muscle pack consists of parallel stacked actuators made-up-of-comprising serially stacked pouches. A pouch contracts linearly under the application of a voltage. **b**, Working mechanism of PELE. We applied a voltage to the muscle packs at a frequency of 0.5 Hz, leading to a cyclic motion of the leg's tip position. **c**, Closed-loop position control of the leg's tip position. The leg's tip position (in relation to the origin depicted in subfigure a) was close loop controlled to precisely track the trajectory of four predefined shapes for a duration-of-20 sseconds. Encoders in each joint translated angle measurements into actuator displacements.

Remark 2-4:

Line 143. How was optimization carried out?

We thank the reviewer for the remark regarding our optimization. Our optimization aimed to reduce the leg system's weight. We designed several versions, iteratively reducing the system's weight, especially for the 3D-printed parts, while considering the rigidity and functionality of the system. The actuator power-to-weight was

increased by downscaling the pouch volume. The attachment mechanism of the actuators to the skeleton was also redesigned using a glass-fiber reinforced film instead of an acrylic plate for weight reduction.

Line 576ff:

The structure was iteratively improved to reduce the weight while considering rigidity and functionality of the system.

Line 605ff:

To reduce the muscle weight, the original 20 mm-long HASEL pouches were miniaturized to 10 mm in length. This modification led to a reduction in the weight of the filled liquid dielectric. Additionally, lightweight glass fibre fiber plates of 2 mm thickness were utilized used to replace the 5 mm thick acrylic boards typically used for connecting the actuator's plastic film and the tendon.

Remark 2-5:

Line 145 What are the actual values of power and weight for these actuators?

We thank the reviewer for their comment. We have now added power-to-weight measurements to the manuscript. We achieved a maximum of 580 W/kg under a 2 N load and 305 W/kg under a 10 N load.

Line 609ff:

Because of these modifications, the muscles achieved a specific energy of 7.4 J/kg (including the masses of tendon connectors and electrical connectors), marking an enhancement overenhancing the original 2.0 J/kg. For the actuator's power output, the maximum specific power increased to 580 W/kg (Extended Data Figure 7) from the original 160 W/kg (40).

We added the Extended Data Figure 6 (see Figure R5 in this response letter) with a detailed caption.

Figure R5: Extended Data Figure 6 | Specific power of the Peano-HASEL artificial muscle. We analyzed the peak specific power of our Peano-HASEL actuators using various weights (100 g, 200 g,

300 g, 400 g, 500 g, 700 g, 1000 g) following an analysis detailed in Kellaris et al. (40). The muscle recorded the highest specific power of 580 W/kg under around 2 N load.

Remark 2-6:

Figure 3B The time axis should be labeled appropriately with units

We thank the reviewer for the remark and have added the unit time for these axes in Fig. 3b (see Figure R6 in this response letter).

Figure R6: **Fig. 3 | The leg performs agile locomotion.** **a**, High jump. The leg jumped 128 mm high from the ground and reached the highest point in 172 ms. **b**, Agile vertical hopping. Top: the leg was driven by an open-loop controller with the periodic input signal. The partial overlap of the activation of the extensor and flexor muscles during the stance phase was used to compensate for dead zones of the muscle (from approx. 0 kV to 3 kV, depending on external loads, the actuators do not respond to voltage signals) and for inertial delay of the leg motion.

Bottom: the leg hopped at 3 Hz on the rubber surface (left) and the slippery surface (right). **c**, Limit cycles for hopping on a slippery surface. The dashed line showed the mean value of the 15 cycles. **d**, Rapid gait motion. Top: the leg was driven by the open-loop force controller with the periodic input signal. Bottom: The leg achieved 5 Hz gait motion and 10 Hz linear motion. **e**, Size comparison of the area enclosed by the leg's foot trajectory at each frequency. The mean and standard deviation were calculated from multiple motion cycles after a steady limit cycle had been reached, ~~see~~ (details in the Methods-section).

Remark 2-7:

Figure 3B How was the timing of the flexor and extensor input signals determined? Clearly the start of each smooth sinusoidal flexor muscle activity partially overlaps with the pulsatile extensor activation, is this to provide stability through an increase in joint stiffness? I presume that the flexor activity is also used to bring the leg back to its starting stance position faster than gravitational acceleration.

We greatly appreciate the reviewer's questions and comment. We used heuristic experimental iterations to determine the signals. Indeed, we intended to flex the leg smoothly and activate the muscle earlier to increase maximum foot height and prepare for landing. The partial overlap was used to compensate for the inertial delay of the system and to mitigate the dead zone of the Peano-HASEL actuators (from approx. 0-3kV, depending on external load, the actuators do not respond to voltage inputs).

We adapted the caption for Fig. 3b:

b, Agile vertical hopping. Top: the leg was driven by an open-loop ~~controller with the periodic input signal~~ force controller with the periodic input signal. The partial overlap of the activation of the extensor and flexor muscles during the stance phase was used to compensate for dead zones of the muscle (from approx. 0 kV to 3 kV, depending on external loads, the actuators do not respond to voltage signals) and for inertial delay of the leg motion. Bottom: the leg hopped at 3 Hz on the rubber surface (left) and the slippery surface (right).

Remark 2-8:

It would be interesting to characterize the work loop properties of the different muscles for comparison with known work loops in jumping animal legs.

We thank the reviewer for this comment. We assume the reviewer's remark mainly concerns Fig. 3b (see Figure R6 in this response letter) on agile vertical hopping. Therefore, we have now measured and reported a work loop analysis for the knee extensor muscle. We tried to design hereby an experiment that, as close as possible, matches typical approaches used in the field of animal biomechanics. We chose the knee extensor muscle, which is dominant in a vertical hopping scenario. The results shown in the Extended Data Figure 5 (see Figure R7 in this response letter) exhibit fast responses of the HASEL muscle to stimulations (applied voltage). Our equipment (310C-LR, Aurora Scientific) allowed us to test the work loops for a range of 2 Hz to 6 Hz; for higher frequencies, the muscle tester could not produce smooth sinusoidal

displacement. We appreciate the reviewer's suggestion to evaluate the dynamic properties of the muscles, which guided us to further insights into their performance.

Figure R7: Extended Data Figure 5 | Work loop analysis of the knee extensor muscle.

a, Experimental setup, based on an 310C-LR, Aurora Scientific, used to evaluate work loops during vertical hopping at different frequencies (shown here are 5 consecutive cycles for each frequency in steady state conditions). The work loops show similar trajectories across different hopping frequencies. The muscle displacement during the hopping experiment was emulated by prescribing a specific sinusoidal displacement signal (8 mm peak-to-peak; equivalent to 5 % actuator strain) for this work loop analysis. The zero-point of the displacement was set as the initial length of the muscle without stimulation (0 kV). b, Impact of stimulation timing on work loop.

We now reference the new Extended Data Figure 5 (see Figure R7 in this response letter) in line 154ff:

The leg's low inertia, combined with the fast response (Extended Data Figure 5) and high power-to-weight ratio (Extended Data Figure 6) of the muscles, enabled the leg to achieve leg's agile motions.

Remark 2-9:

Figure 4, panel a needs actual time units indicated

We thank the reviewer for catching this oversight. We have now added the time unit for these axes to Fig. 4a (see Figure R8 in this response letter).

a Versatile locomotion over varying terrain with solely open-loop muscle force control

Figure R8: The leg is inherently adaptive using open-loop force control. a, Versatile locomotion over varying terrain with solely open-loop force control.

Remark 2-10:

The stiffness of the muscle and their attachment points should be stated, it would be helpful to have passive and active work loop information (i.e. stress-strain relationships) of the actuator itself and in combination with jointed arm.

We thank the reviewer for the suggestions which allowed us to enhance both the clarity of our writing and to add additional results to explain our work. The attachment points are now stated in the Extended Data Figure 1. We have also measured the muscle and leg stiffness and plotted their work loops (Extended Data Figure 7). The measurement showed that the stiffness of both the individual muscle and the whole leg system increases when the applied voltage increases. These results enhance our argument of tunable stiffness, and we thank the reviewer for suggesting these new experiments.

We adapted the manuscript line 128ff:

For PELE, we attach one end of the actuator rigidly to the leg, and the tendon on the other end to the shank after the respective joint (Fig. 2a2a, Extended Data Figure 1). Each tendon has a non-linear moment arm transmission for suitable angle-torque profiles of the joints (Extended Data Figure 2).

Line 231ff:

This inherent soft-landing feature is enabled by the muscle's inherently tunable back drivability (Extended Data Figure 7) and eliminates the need for complex computational controllers (45) to adjust the leg's stiffness.

Figure R9: Extended Data Figure 1 | **Leg geometry.** The leg's basic geometry with measurements of the lever arms and angles is required to simulate its kinematics. The actuators are fixed to one end of the carbon fiber "bones" using a 3D-printed attachment. The distance to the bone's central axis is at least 12 mm. Actuators are stacked one atop the other, increasing that distance. A tendon made from a fishing line connects the other end to the lever arm.

Figure R10: Extended Data Figure 7 | **Stiffness of the HASEL muscle and PELE changes proportionally to the applied voltage.** **a**, Experimental setup, based on an 310C-LR, Aurora Scientific, used to evaluate the stretching stiffness of the Peano-HASEL artificial muscle at different voltage levels. The slope angles of the trajectories represent the muscle stiffness. Higher voltage causes higher stiffness of the muscle. **b**, Experimental setup, based on an 310C-LR, Aurora Scientific, used to evaluate the compression stiffness of the leg. Again, higher applied voltage causes higher leg stiffness.

Remark 2-11:

It would be a good idea to have a more recent reference for the energetic cost of animal locomotion, the one used here was from 1975. This should also be confined to terrestrial legged locomotion. Eg Halsey, L.G., White, C.R. Terrestrial locomotion energy costs vary considerably between species: no evidence that this is explained by rate of leg force production or ecology. *Sci Rep* 9, 656 (2019). <https://doi.org/10.1038/s41598-018-36565-z>

Pontzer H. A unified theory for the energy cost of legged locomotion. *Biol Lett.* 2016 Feb;12(2):20150935. doi: 10.1098/rsbl.2015.0935. PMID: 26911339; PMCID: PMC4780550.

We agree with the reviewer and appreciate the suggestions for more up-to-date literature on the energetic cost of animal locomotion. We acknowledge the difficulty of defining a uniform COT line for land animals; nevertheless, we have adapted the baseline in Fig. 5b (Figure R3 in this response letter) based on the suggested reference (Pontzer, 2016), to provide a reference for overall comparison.

Remark 2-12:

I am unclear about the cost of transport estimates. Although the total cost and net cost for other robots are included in figure 5, only the net cost for this robot are shown. The net cost of transport is calculated from the electrical energy supplied from the power supply to the leg. You should include an estimate of power lost in the high-voltage conversion because this will affect the energy storage capability and untethered lifetime of the robot.

We thank the reviewer for their remark and added a sentence on the estimated power loss in high voltage power supplies.

In this work, we used a PolyK PK-HVA1005 power supply, a multi-kilogram precision benchtop power supply unsuitable for untethered operation. This power supply is not optimized for efficiency and, therefore, is not included in the COT calculation. Please see Remark 1-8 for details on the COT calculation.

We can typically assume that the primary energy loss in high-voltage power supplies happens in the DC-DC conversion. As we only turn the power supply on and off for jumping, the switching circuits can be kept very simple and are neglected for this approximation. The DC-DC conversion efficiency of small-scale HV-power converters is generally above 75%, depending on output power (Xie, 2020). Lodh et al. have shown an ultra-high gain converter for driving HASEL actuators with an efficiency of 78.2% (Lodh, 2023). Commercially available ultra-high gain converters with 10kV output have efficiencies between 75% to 83% at full load (PICO Electronics, 2023). A comment on a typical level of energy loss was added to the manuscript.

For more complex control circuits needed for high-frequency switching or bipolar switching, the energy losses in the switching circuit should also be considered. Improving the efficiency of these circuits is a key focus of ongoing research efforts.

Xie et al. 2020, IEEE Xplore. <https://doi.org/10.1109/APEC39645.2020.9124193>

Lodh et al. 2023, Biomimetics. <https://doi.org/10.3390/biomimetics8010053>
PICO Electronics 2023, <https://www.picoelectronics.com/node/13283>

Line 264ff:

The results indicate that the leg on a boom arm achieved a desirable low cost of transport (~~COT, Equation 7~~), reflecting efficient movement with low energy consumption. Notably, the leg achieved a minimum ~~COT of 0.69 at a reduced weight of 198 g, and a net COT of 0.73 at the original weight of 225 g, and a net COT of 0.69 and 0.83 at a reduced weight of 198 g and an increased weight of 391 g, respectively.~~ Intriguingly, as the weight exceeded 225 g, the locomotion type that yielded the minimum net COT transitioned from hopping to crawling. Compared to other legged ~~robots, the leg systems (Extended Data Table 3), PELE~~ exhibited remarkably ~~favourable~~favorable COT values within this range of robotic mass (Fig. 5b, Extended Data Table 2). The net COT calculation does, however, not include the DC-DC high voltage conversion efficiency in the driving electronics of typically around 75 % (51,52). These findings suggest that the electrohydraulic muscle system has the potential for enabling highly efficient locomotion in untethered legged robots.

Remark 2-13:

How was the crawling gait to defined? There is no definition or demonstration of crawling.

We defined the “crawl gait” as gait without flight phase. This gait can be observed in the efficiency video in the time frames starting at 00:28 and ending at 00:35.

We adapted the main manuscript to clarify what “crawl gait” means.

Line 239ff:

We specifically investigated two different types of gaits, hopping gait and crawling gait. We defined crawling gait as hopping without a flight phase (it can be seen in Supplementary Video 5 from 00:28 to 00:35).

Remark 2-14:

The energy efficiency of maintained squatting posture is an exciting finding and the simple open-loop obstacle detection system is also an appealing feature of this robot design. Of course it is currently limited to a simple mode switching control feature but I'm sure this could be expanded to include additional responses or gates dependent on the proprioceptive feedback.

We agree with the reviewer and are thankful that the reviewer appreciates our findings. This is an extremely rich area of exciting research that goes well beyond the scope of this paper and justifies a separate future paper. We are looking forward to future studies where we will see systems that measure muscle length in real-time and use those measurements for proprioceptive feedback enabling more bio-inspired gait control methods.

Remark 2-15:

It is pleasing to see some of the limitations of this creative design honestly discussed. Although it's not necessary to include here, the other limitation for practical applications is obviously the high voltages necessary for the HASEL actuators. I suspect that this danger could be mitigated by appropriate packaging.

We thank the reviewer for the remark and agree that sharing the limitations of a system is an essential part of the scientific conversation. Regarding the high voltages necessary, we have included additional explanations concerning high voltage safety. Importantly, we would like to point out that high voltages alone are not necessarily dangerous, only in case this is paired with high currents. HASEL actuators operate at high electric fields (high voltages) and typically low currents which are well within established safety standards. Ironically, for an actuator that outputs 10W of mechanical power, an actuator operated at 10kV and 1mA is safer compared to an actuator that operates at 1kV and 10mA [Rothemund et al., 2021].

To ensure safe operation in practical applications, typical safety measures involve restricting current flow to safe levels [Standard IEC-60479-1] implementing mechanisms that detect electrical breakdowns and trigger fast shutoff circuitry for driving electronics, limiting maximum capacitance/energy stored in the system according to safety standards (Standard NFPA 70E), and packaging/encapsulation of the entire system to prevent any exposed electrical connections. If needed, reducing the maximum capacitance of the system can be achieved by compartmentalizing the system into smaller subsystems, and connecting these subsystems via fuses that disconnect them in case of electrical breakdowns.

We have addressed the high voltage safety of the HASEL technology in more detail in a recent review paper [Rothemund et al., 2021], that we now also reference in this context in our manuscript, to provide the readership with additional resources on this topic.

References:

Standard IEC-60479-1 <https://webstore.iec.ch/publication/62980>

Rothemund, et al., Advanced Materials, 2021,
<https://doi.org/10.1002/adma.202003375>

Standard NFPA 70E, <https://www.nfpa.org/codes-and-standards/nfpa-70e-standard-development/70e>

We changed our manuscript as follows to address high-voltage safety:

Line 397ff:

(v) Currently, the electrodes of the actuator are open to the environment and at a high voltage potential. High voltages alone are not necessarily dangerous, only when paired with high currents. HASEL actuators operate at high electric fields (high voltages) and at low currents, which are within established safety standards, unless we perform highly dynamic motions like jumping. To ensure safe operation in practical applications outside a controlled laboratory environment, additional safety measures would be required. These could involve (a) implementing mechanisms that detect

electrical breakdowns as well as fault currents, and then trigger fast shutoff circuitry for driving electronics, and (b) limiting maximum capacitance/energy stored in the actuator system. Additionally, (c) introducing electrical insulation layers (47) to prevent physical contact with the electrode and therefore electrical shock can help mitigate such risks (58)

Response to Reviewer #4

Remark 4-1:

1. The PELE is attached to a boom for locomotion and jump tests. In Fig. 5b, the PELE is compared to multiple-legged robots and land animals which do not use a boom for support and are instead self-standing. This makes the COT comparison feel a little incomplete. At the least, this discrepancy should be noted in the main text of the manuscript. Additionally, the use of the boom support should be mentioned in the abstract for clarity.

We thank the reviewer for the remark and have now updated the manuscript to describe the system as “a robotic leg on a boom arm”. In addition, we have now added the minimum cost of transport for legs on boom arms to our COT overview in Figure 5b (see Figure R3 in this response letter).

Change to the abstract:

Line 25f:

~~Our legged system~~ Our leg is mounted on a boom arm and can adaptively traverseshop on varying terrain in an energy-efficient yet agile manner and detects. It can also detect obstacles through capacitive self-sensing.

Change to the main manuscript:

Line 390ff:

(ii) So far, we have only shown a single 2-DOF leg mounted on a boom, which limits the leg’s locomotion to a circular path. Hexapedal, quadrupedal, and bipedal systems must be investigated to fully characterize the potential of this design architecture.
(iii) We only compared the energy efficiency for squatting. It would be important to extend the investigation into other types of motion, for example, continuously fast actuation like used for running.

Remark 4-2:

2. Please mention either in the Fig. 5 caption or in the main text whether the COT for PELE is a net or total COT measurement.

We thank the reviewer for the remark. The COT calculated for PELE was net COT, which we have now clarified in the manuscript and in Fig. 5b (see Figure R3 in this response letter).

Line 265ff:

Notably, the leg achieved a minimum COT of 0.69 at a reduced weight of 198 g, and a net COT of 0.73 at the original weight of 225 g, and a net COT of 0.69 and 0.83 at a reduced weight of 198 g and an increased weight of 391 g, respectively. Intriguingly, as the weight exceeded 225 g, the locomotion type that yielded the minimum net COT transitioned from hopping to crawling. Compared to other legged robots, the leg systems (Extended Data Table 3), PELE exhibited remarkably favourable favorable COT values within this range of robotic mass (Fig. 5b, Extended Data Table 2). The net COT calculation does, however, not include the DC-DC high voltage conversion efficiency in the driving electronics of typically around 75 % (51,52).

Remark 4-3:

3. Please include a discussion of the expected cycles to failure as well as the expected durability of the actuators (Durability in terms of reactions to external contact and punctures).

We greatly thank the reviewer for their remark on the expected durability. We have now commented on the puncture resistance and added a paragraph on the lifetime and durability of HASEL actuators. In the real-world application of a HASEL-driven system, an enclosure or puncture-proof layer (e.g., aramid fibers) around the actuator would be necessary. In general, a lifetime assessment of “one-off” prototypes is challenging to conduct. Especially when human error (e.g., dust contamination) in the production is a major contributing factor to reduced lifetimes. High-quality HASEL actuators have been tested for lifetimes exceeding 10^6 cycles [Acome, 2018; Morton, 2021].

Reference:

Acome, et al., 2018, *Science*, <https://doi.org/10.1126/science.aao6139>

Morton, 2021, <https://www.artimusrobotics.com/post/lifetime-of-hasel-actuators>

Line 381ff:

The HASEL actuators have a lifetime of thousands up to exceeding millions of cycles (57), depending on the design and demand of a particular application. In the current design, the actuators are not shielded from contact with the environment. In a real-world application of a HASEL-driven system, an enclosure or puncture-proof layer (for example, aramid fibers) around the actuator would be necessary.

Remark 4-4:

4. Please provide an explanation or hypothesis as to why the gait range of motion increases and then decreases as the gait frequency increases in Fig. 3e.

We thank the reviewer for their remark. We believe this behavior is in line with our expectations. The operation frequency matches the natural frequency of the leg system, resonance increases the range of motion, and the moment of inertia around the hip joint reduces the range of motion of the hip joint at higher frequencies. We also observed stable and high jumps at around 2.5Hz to 3Hz in the vertical hopping tests. We added these findings in the according section.

Line 190ff:

Additionally, the foot's range of motion changed for different operation frequencies, with a maximum of 3 Hz (Fig. 3e), which matched the natural frequency of the leg system. For 3 Hz, the leg achieved a jump height of 80 mm and was stable in vertical hopping.

Remark 4-5:

5. The HASEL acronym should be first spelled out on line 69 rather than line 108.

Thank you, we have adapted the manuscript accordingly.

Line 69f:

We test the hypothesis by introducing PELE, an efficient and agile Peano-HASEL (hydraulically amplified self-healing electrostatic actuator) driven ~~Leg~~ with 2-DOF.

Remark 4-6:

6. Comparisons with PELE were done with a DC-motor-driven leg of comparable size and weight. Care should be taken especially during the energy efficiency comparisons as the DC-motor leg likely can support a higher mass payload than PELE, such as shown in the DC-motor systems shown in Fig. 5b. For a more apt comparison the DC-motor leg should have been undersized in terms of size and/or weight and should instead be sized to match the mass payload capacity of the PELE. Alternatively, if the DC motor were controlled via the drive modules in a way such as to match this payload capacity that would also be acceptable. If that is the case in the current manuscript it is not clear.

7. Continuing from 6., when stating that PELE required ~0.6% of the energy of a comparable DC-motor-driven leg please instead compare it with a DC-motor-driven leg that either matches the total mass payload capacity of the PELE or show/state that only energy as needed is supplied to the DC motors until it matches the payload capacity of PELE, without excess.

We thank the reviewer for this remark. Choosing the correct motors for a fair comparison requires considering many factors, including mass, payload/torque output, system complexity, and adaptability/back drivability

For the DC-motor-driven leg, we selected a direct-drive approach to match the mass, back-drivability, bandwidth, torque production and simplicity of PELE's actuators. This design choice brings versatility for different tasks, which, in principle, also matches PELE, where a single design achieves different tasks. For the DC-motor (T-MOTOR, MN4006), the selection criteria were its weight and torque output to be comparable to that of the Peano-HASEL stacks, which met the quasi-static joint torque requirements over the entire range of joint angles. To demonstrate the torque relationships, we added Fig. 5f (see **Error! Reference source not found.** in this response letter), which compares the torque values of the HASEL muscle to the DC motor and the required quasi-static torque. For a substantial portion of the range of the knee angles, the Peano HASEL has an even higher torque output than the DC motor.

Achieving precise matching of "the total mass payload capacity" is challenging due to variations in how maximum forces are quantified. PELE's muscles reduce the force with increasing strain. The blocking force is typically defined as the force at which the muscle no longer moves. The actual usable force range is much lower than the blocking force of the muscle due to the shape of the force/strain curve. In contrast, the DC leg has constant torque. A leg's highest torque is required when it is in its lowest squatting position. If the DC motor can overcome this initial highest torque, a leg powered by a DC motor can complete the whole movement. We added Fig. 5f (see **Error! Reference source not found.** in this response letter) to the main manuscript to better explain to the reader the torque relationships and comparison between the DC-motor leg and PELE.

For additional information on the energy efficiency, please also refer to our answer to Reviewer #1, Remark 1-7 and to Fig. 5 (see **Error! Reference source not found.** in this response letter).

Remark 4-7:

8. The claim that all electromagnetic leg designs without closed-loop control would fail to traverse grass, sand, gravel, pebbles, and large rocks is quite bold. This claim needs to be substantiated more in the text.

We appreciate your critical feedback on this matter, which has helped us improve the accuracy of our claims in the manuscript. We agree with the reviewer and remove these claims because they were based on preliminary results only.

Abstract, line 25ff:

~~Our legged system~~ Our leg is mounted on a boom arm and can adaptively traverse ~~hop on~~ varying terrain in an energy-efficient yet agile manner and detects. It can also detect obstacles through capacitive self-sensing. The leg performs powerful and agile gait motions beyond ~~5 Hz~~ 5 Hz and high jumps ~~of up to 40% body %~~ of the leg height. ~~While an electromagnetic leg design without closed-loop control would fail, our~~ Our leg's tuneable ~~tunable~~ stiffness and inherent adaptability allow it to ~~traverse~~ hop over grass, sand, gravel, pebbles, and large rocks using only open-loop force control.

Main manuscript, line 87ff:

Moreover, the musculoskeletal design and the electrohydraulic actuators of PELE enabled an open-loop force controller via only regulating applied voltage to manipulate and locomote the leg.

To conclude, we hope that the answers above and the expanded and revised version of our manuscript satisfy your requests. Again, we thank you for your insightful comments, which have greatly helped us to improve the quality of our manuscript.

Best Regards,

Authors

REVIEWERS' COMMENTS

Reviewer #1 (Remarks to the Author):

I am happy to say that my comments have almost all been fully addressed. I find the manuscript much improved. I recommend publishing as is

Reviewer #4 (Remarks to the Author):

The authors have sufficiently addressed all of my concerns with the revised manuscript. I thank them for their effort.